



# Convective uplift of pollution from the Sichuan basin into the Asian monsoon anticyclone during the StratoClim aircraft campaign

Keun-Ok Lee[1], Brice Barret[1], Eric L. Flochmoën[1], Pierre Tulet[2], Silvia Bucci[3], Marc von Hobe[4], Corinna Kloss[4,5], Bernard Legras[3], Maud Leriche[1,8], Bastien Sauvage[1], Fabrizio Ravegnani[6], Alexey Ulanovsky[7]

[1]Laboratoire d'Aérologie, Université de Toulouse, CNRS, UPS, Toulouse, France
[2]Laboratoire de l'Atmosphère et des Cyclones, UMR8105, CNRS, Météo-France, Université de la Réunion, Saint-Denis de La Réunion, France
[3]Laboratoire de Météorologie Dynamique, UMR CNRS 8539, IPSL ENS-PSL/Sorbonne Université/Ecole Polytechnique/Ecole des Ponts Paris Tech, Paris, France
[4]Institute for Energy and Climate Research (IEK-7), Forschungszentrum Jülich GmbH, 52425 Jülich, Germany
[5]LPC2E, CNRS/ Université d'Orléans, France
[6]Institute for Atmospheric Sciences and Climate of the National Research Council (CNR-ISAC), Bologna, Italy
[7]Central Aerological Observatory (CAO), Moscow, Russia
[8]Centre pour l'étude et la simulation du climat à l'échelle régionale, Université du Québec à Montréal, Montréal, Canada

*Correspondence to*: Keun-Ok Lee (keun-ok.lee@aero.obs-mip.fr) and Brice Barret (brice.barret@aero.obs-mip.fr)

**Abstract.** The StratoClim airborne campaign took place in Nepal from 27 July to 10 August 2017 to document the physical and chemical properties of the South Asian Upper Troposphere Lower Stratosphere (UTLS) during the Asian Summer Monsoon (ASM). In the present paper, simulations with the Meso-NH cloud-chemistry model at a horizontal resolution of

15 km are performed over the Asian region to characterize the impact of monsoon deep convection on the composition of Asian Monsoon Anticyclone (AMA) and on the formation of the Asian Tropopause Aerosol Layer (ATAL) during the StratoClim campaign. StratoClim took place during a break phase of the monsoon with an intense convective activity over south China and Sichuan. Comparisons between Brightness Temperature (BT) at 10.8 microns observed by satellite sensors and simulated by Meso-NH highlight the ability of the model to correctly reproduce the life cycle of deep convective clouds.

Comparison between CO and O$_3$ concentrations from Meso-NH and airborne observations (StratoClim and IAGOS) demonstrates that the model captures most of the observed variabilities. Nevertheless, for both gases, the model tends to overestimate the concentrations and misses some thin CO plumes related to local convective events probably because of a too coarse resolution, but the convective uplift of pollution is very well captured by the model. We have therefore focused on the impact of Sichuan convection on the AMA composition. A dedicated sensitivity simulation showed that the 7 August

convective event brought large amounts of CO deep into the AMA and even across the 380 K isentropic level located at 17.8 km. This Sichuan contribution enhanced the CO concentration by ~15 % to reach more than 180 ppbv over a large area around 15 km height. Noteworthy, Meso-NH captures the impact of the diluted Sichuan plume on the CO concentration during a StratoClim flight south of Kathmandu highlighting its ability to reproduce the transport pathway of Sichuan pollution. According to the model, primary organic aerosol and black carbon particles originating from Sichuan are





transported following the same pathway as CO. The large particles are heavily scavenged within the precipitating part of the convective clouds but remain the most important contributor to the particle mass in the AMA. Over the whole AMA region, the 7 August convective event resulted in a 0.5% increase of CO over the 10−20 km range that lasted about 2 days. The impact of pollution uplift from three regions (India, China and Sichuan) averaged over the first 10 days of August has also been evaluated with sensitivity simulations. Even during this monsoon break phase, the results confirm the predominant role

of India relative to China with respective contributions of 11 and 7 % to CO in the 10–15 km layer. Moreover, during this period a large part (35 %) of the Chinese contribution comes from the Sichuan basin alone.

# 1 Introduction

Deep convection plays a key role in venting chemical constituents from the Boundary Layer (BL) to the Upper Troposphere-Lower Stratosphere (UTLS), where they have a significant chemical and radiative impact (Mason and Anderson, 1963;

Dickerson et al., 1987; Randel and Park, 2006; Su et al., 2011; Fadnavis et al., 2013; Gu et al., 2016). With updraft velocities that can largely exceed 10 m s$^{-1}$, vertical uplift from deep convection can occur in duration ranging from minutes to hours (Markowski and Richardson, 2010).

The Asian Summer Monsoon (ASM) circulation covers large parts of south and East Asia from the tropics to the subtropics. It consists of a cyclonic flow and convergence in the lower troposphere associated with a strong anticyclonic

circulation and divergence in the UTLS, the Asian Monsoon Anticyclone (AMA). This circulation is coupled with persistent deep convection over the south Asian region during summer (June to September) (Hoskins and Rodwell, 1995). The tropopause height above the ASM is relatively high (16−17.5 km) and the AMA extends into the lower stratosphere spanning from around 200 hPa to 70 hPa (12−18.5 km above sea level), i.e. approximately the whole UTLS (Highwood and Hoskins, 1998; Randel and Park, 2006).

Due to the strength and size of the AMA, which spans the region from the Pacific to the Mediterranean, the influence of the monsoon on the UTLS composition is a significant contribution for the global budget. In the past 15 years, numerous studies based on satellite data have shown that the AMA had a strong BL composition signature. For instance, using Microwave Limb Sounder (MLS) data, Wright et al., (2011) have documented relatively high water vapour mixing ratios (about 4.2–4.5 ppmv) and Atmospheric Infrared Sounder (AIRS) data show low Ozone (O$_3$) concentrations (Randel et al.,

2001) within the AMA. With their high and unprecedented UTLS sensitivity, the MLS carbon monoxide (CO) observations clearly allowed the detection of the BL signature in the AMA (Park et al., 2004; Li et al., 2005; Randel and Park, 2006; Barret et al., 2008; Randel, 2010). More recently, observations from the CALIPSO spaceborne lidar evidenced the Asian Tropopause Aerosol Layer (ATAL) distributed around 16 km within the AMA (Vernier et al., 2015).

Most of the studies based on chemistry transport simulations have demonstrated that the BL pollution uplifted to the

AMA was mostly from Indian or South Asian sources (Park et al., 2009; Barret et al., 2016). Aerosols in the UTLS have much longer residence times than in the lower troposphere and can therefore impact atmospheric chemistry and the Earth's



climate with large spatial and temporal coverage (Rasch et al., 2008). Eastern Asia and China are highly polluted but less impacted by monsoon deep convection and therefore contribute less to the feeding of BL pollution into the UTLS. Nevertheless, deep convection is characterised by an important inter-annual variability. The active phase of the StratoClim

(Stratospheric and upper tropospheric processes for better Climate predictions, www.stratoclim.org) airborne campaign took place during a monsoon break phase with a convective activity particularly strong over the Sichuan Basin compared to Nepal and northern India (Lee et al., 2019; Bucci et al., 2020).

StratoClim aims at improving our knowledge about the key microphysical, chemical and dynamical processes that control the composition of the AMA and of the ATAL and the impact of the ASM on the hydration of the stratosphere. During the

campaign (from 27/07/2017 to 10/08/2017), eight dedicated flights were successfully carried out. Using StratoClim in situ measurements and convection-permitting numerical simulations, Lee et al. (2019) investigated the stratospheric hydration documented during flight #7. They reported that the strong orographically-triggered overshooting convection occurring in the Sichuan basin (Figure 1) transported more than 6 kt of water vapour across the tropopause. Due to strong UTLS easterly winds south of the AMA, a large amount of water vapour ($\geq$ 5.5 ppmv) has been transported over Nepal even though a large

part of the uplifted moisture has been lost due to ice formation and sedimentation.

Sichuan is a highly populated (83 million according to 2017 estimation) and industrialised region with large cities such as Chengdu, Mianyang, and Nanchong, resulting in large amounts of pollutants stored in the basin at the mountain foothills. CO emissions from the MACCity inventory (https://eccad3.sedoo.fr/) (Figure 2) show the isolated large CO fluxes in this region (red box). We can therefore hypothesize that the unusual over-shooting convection documented by Lee et al. (2019) resulted

in the uplift of large amounts of pollutants into the AMA and its westward transport by the AMA easterlies. We therefore focused our investigation on the impact of the unusual Sichuan overshooting convection on the AMA composition during the StratoClim campaign. This is achieved by combining fine-scale cloud-chemistry simulations with the unprecedented wealth of measurement data.

Detailed descriptions of the observations and of the model simulations are given in section 2. Section 3 presents a

validation of the simulated convective clouds with satellite observations and of the simulated UTLS composition with airborne in situ measurements. In section 4, we investigate the pollution uplift to the AMA by the strongest overshooting convective event in the Sichuan Basin. Section 5 is dedicated to quantifying the impact of broader regional sources on the UTLS composition. Finally, a summary and discussion of the findings of the present study are provided in section 6.



## 2 Data and method

### 2.1 Airborne observations: StratoClim and IAGOS

#### 2.1.1 StratoClim measurement

We will use data from M55-Geophysica flights #5, #6, #7, and #8, which took place from Kathmandu in Nepal (Table 1; for the tracks see red lines in Fig. 1). During those flights, the AMICA (Airborne Mid Infrared Cavity enhanced Absorption spectrometer) and FOZAN-II (Fast OZone ANalyzer) instruments measured the CO and $O_3$ concentrations respectively. During 03:00−07:25 UTC on 4 August for flight #5, during 07:30−11:30 UTC on 6 August for flight #6, during 04:30−06:50 UTC on 8 August for flight #7 (exceptionally no $O_3$ measurement), and during 08:40−12:30 UTC on 10 August for flight #8.

Carbon monoxide was measured by the AMICA instrument (Kloss et al., *in preparation*) placed on top of the Geophysica aircraft. AMICA employs Integrated Cavity Output Spectroscopy (ICOS, O'Keefe et al., 1999) to measure various trace gases in the mid infrared region. CO mixing ratios were retrieved from observed spectra using a transition at 2050.90 $cm^{-1}$ with line parameters taken from the HITRAN 2012 database (Rothman et al., 2013) and no further calibration parameters. Accuracy was tested for a range of mixing ratios (30–5000 ppb) prepared from a 5 ±0.05 ppm CO standard (AirProducts) and is estimated to be better than 5 % (taking into account the uncertainty of the standard, upper limits for impurities in the dilution gas, and uncertainties in the MFC flows used for dilution. In this study, AMICA CO data are used at 10 s time resolution and have a 1-sigma precision of ~20 ppb that was mainly limited by electrical noise on the observed spectra.

The FOZAN-II is a chemiluminescence sensor for $O_3$ monitoring at 1 Hz time resolution. It was developed by the Central Aerological Observatory, Russia, and Institute of Atmospheric Science and Climate, Italy (Yushkov et al., 1999; Ulanovsky et al., 2001) and is jointly operated aboard Geophysica by scientist from the two Institutes. FOZAN-II is a two-channel solid state chemiluminescent instrument featuring a sensor based on Coumarin 307 dye on a cellulose-acetate-based substrate, and is equipped with a high accuracy ozone generator for periodic calibration of each channel every 15 minutes during the flight ensuring an accuracy better than 10 ppb and precision of 8 %. The measured concentrations range is 10–500 $\mu g\ m^{-3}$; operating temperature range is –95 to +40 °C; and the operating pressure range is 1100–30 mbar (about 0–22 km). Instrument was calibrated at ground before and after each flight by means of ozone generator and reference UV-absorption $O_3$ monitor (Dasibi 1008-PC).

#### 2.1.2 IAGOS measurement

A commercial aircraft equipped with a IAGOS (In-service Aircraft for a Global Observing System, https://www.iagos.org/) instrumental package flew back and forth between Frankfurt in Germany and Madras in India (for the tracks, see the red solid and dashed lines in Fig. 1) from 09:18 UTC on 5 August to 05:22 on 6 August 2017 having a short break at Madras from 18:20 UTC to 20:20 UTC on 5 August 2017. In situ sensors aboard the aircraft measured the CO and $O_3$ concentration every 30 s and 4 s, respectively. The CO analyser (Nedelec et al., 2003) is an improved version of a commercial Model



48CTL from Thermo Environmental Instruments, based on the Gas Filter Correlation principle of infrared absorption by the 4.67 µm fundamental vibration-rotation band of CO. The Model 48CTL is qualified by U.S. EPA designated Method (EQSA-0486-060). The precision specification of the commercial instrument is within ±5 %.

The IAGOS $O_3$ analyser (Thouret et al., 1998) is a dual-beam UV absorption instrument (Thermo-Electron, model 49-103). The response time is 4 s, and the concentration is automatically corrected for pressure and temperature influences. This instrument archives a precision of ±2 %, an accuracy of 1 ppbv, and a minimum detectable concentration of 2 ppbv. This instrument provides high stability making the measurements accurate and reliable over long time periods.

**2.2 Spaceborne observation**

In order to document deep convective clouds we used calibrated thermal infrared brightness temperature (BT) in the infrared window from geostationary satellites. In order to correctly cover the region between South and East Asia, BT acquired by the Meteosat Second Generation satellite (MSG)/Spinning Enhanced Visible and Infrared Imager (SEVIRI) sensor were merged with BT acquired by the Advanced Himawari Imager (AHI) sensor aboard Himawari-8 that acquires images every 10 minutes. MSG-SEVIRI, since July 2016, is centred at 41.5°E while Himawari-8 field of view is centred at 140°E. Together, merged at 90°E, they cover the whole region of interest. Data are projected onto a 0.1° grid using a closest neighbour interpolation and sampled every hour. The two satellites are matched at the longitude 90°E. Cold BT below 230 K are indicative of high cloud tops associated with deep convection (e.g. Lee et al., 2016, 2019).

**2.3 High resolution cloud-chemistry simulation**

The coupled cloud-chemistry non-hydrostatic Meso-NH model (Lac et al. 2018) was used to simulate the deep monsoon convection and the AMA composition. The simulations are performed within a large domain covering India and China (Fig. 1, 8.8−40.9°N, 58−122.9°E) therefore encompassing the StratoClim flight tracks and the deep convective clouds over the Sichuan Basin with a 15 km horizontal resolution (~7 million grid points). The vertical grid has 64 stretched levels (Gal-Chen and Somerville, 1975) with a resolution of 100 m close to the surface stretched to 450 m in the UTLS up to 24.2 km. The control simulation (referred to as CNTL in the following) covered the period from 00:00 UTC on 27 July 2017 to 00:00 UTC on 15 August 2017 with three-dimensional outputs every 3 h, assigning first few days as a spin up time.

The initial and lateral boundary conditions for meteorology are provided by the operational European Centre for Medium-Range Weather Forecasts (ECMWF) analyses every 6 h. The initial and boundary conditions for gaseous chemical species including inorganic nitrogen species, e.g. $O_3$, CO, $SO_2$, $NH_3$, NMVOCs (non-methane Volatile Organic Compounds), primary (e.g. black carbon (BC), primary organic aerosols (POA)) and secondary (e.g. inorganic, secondary organic aerosols (SOA)) aerosol species are taken from MOZART-4 (Model for Ozone and Related chemical Tracers, version 4) (Emmons et al., 2010) driven by meteorology from NCEP (National Centres for Environment Prediction). Boundary chemical fields are forced every 6 h. Surface emissions of atmospheric compounds are taken from the ECCAD (Emissions of atmospheric Compounds & Compilation of Ancillary Data) database (https://eccad.aeris-data.fr) at a 0.5° horizontal grid spacing. For



anthropogenic and biomass burning emissions, we have used the MACCity inventory (MACC/CityZEN EU projects) (Van der Werf et al., 2006; Lamarque et al., 2010; Granier et al., 2011; Diehl et al., 2012) corresponding to August 2017. MACCity
provides "off-line" emissions for gases such as alkanes, alkenes, alcohols, aldehydes, ketones and aromatics, lumped into 21 species, and for primary aerosol species. The MEGAN v2 model (Model of Emissions of Gases and Aerosols from Nature) (Guenther et al., 2006) provides "off-line" monthly net biogenic emissions of gases ($NO_x$ and VOCs) and aerosols. Finally, the monthly GFEDv3 (Global Fire Emissions Database, version 3) inventory (Van der Werf et al., 2010) was used for biomass burning emissions (e.g. CO, NMVOCs, BC, POA).

For gas phase chemistry, we used the ReLACS2 scheme (Regional Lumped Atmospheric Chemical Scheme 2) as described by Suhre et al. (1998) and Tulet et al. (2003). ReLACS2 (Tulet et al., 2006) is derived from a reduction of the CACM scheme (Griffin et al., 2002). It includes 82 prognostic gaseous chemical species and 363 reactions enabling the formation of SOA precursors to be addressed. The processes controlling the aerosol population, emissions, nucleation, coagulation, condensation, dry deposition, sedimentation, diffusive transport, and wet-deposition are modelled by the
ORILAM scheme (Organic-Inorganic Lognormal Aerosol Model; Tulet et al., 2005, 2006, 2010). Two lognormal modes of particles are considered, mode #1 (i.e. Aitken mode) of smaller particles with initial mean radius of 0.036 μm and standard deviation (σ) of 1.86, and mode #2 (i.e. accumulation mode) of larger particle with initial mean radius of 0.385 μm and σ of 1.29. The gas to particle conversion for inorganic species is handled by the EQSAM model (Equilibrium Simplified Aerosol Model; Metzger et al., 2002). To simulate SOA formation, the partitioning of the low volatility organic species between the
gas and aerosol phases is based on the thermodynamic equilibrium scheme MPMPO (Model to Predict the Multiphase Partitioning of Organics; Griffin et al., 2003, 2005). The dry deposition of chemical species is treated according to the resistance concept of Wesely (1989) in the SURFEX model, which treats all surface processes (Masson et al., 2013). The deposition depends on the boundary layer turbulence and on the molecular diffusion, which lead gases and particles to effective surface deposition.

Deep convection is parameterised following the Kain-Fritsch-Bechtold scheme (Bechtold et al., 2001). The one-moment bulk microphysical scheme (Pinty and Jabouille, 1998) governs the equations of six water categories (water vapour, cloud water, rainwater, pristine ice, snow and graupel). For each particle type, the size follows a generalized Gamma distribution while power-law relationships allow the mass and fall speed to be linked to the diameters. Except for cloud droplets, each condensed water species has a nonzero fall speed. The turbulence parameterisation is based on a 1.5-order closure (Cuxart et
al., 2000) of the turbulent kinetic energy equation and uses the Bougeault and Lacarrere (1989) mixing length. Momentum variables are transported with the weighted essentially non-oscillatory (WENO) scheme (Shu and Osher, 1988) to transport momentum variables while other variables are transported with the piecewise parabolic method (PPM) scheme (Colella and Woodward, 1984). The SEVIRI/MSG BTs are compared to synthetic BTs computed offline using the Radiative Transfer for TIROS Operational Vertical Sounder (RTTOV) code version 11.3 (Saunders et al., 2013) from the simulation outputs
(Chaboureau et al., 2008).



# 3 Evaluation of CNTL experiment

## 3.1 Clouds and deep convection

Deep convective clouds (BT ≤ 210 K) are detected over the Sichuan Basin at 12:00 UTC on 7 August 2017 by combined observations from the MSG/SEVIRI and Himawari sensors (box, Fig. 3a). Between 12:00 UTC and 18:00 UTC, multiple
deep convective cells develop, and the area of low BT values (≤ 210 K) covers most of the Sichuan Basin at 18:00 UTC (Fig. 3c). It is worth noting the spatial coincidence of deep convection and large Sichuan Basin CO emissions (red box, Fig. 2). From 18:00 UTC (Figs. 3e, g) on the convective systems gradually stretch horizontally with the strong UTLS easterlies. During Geophysica flight #7 (8 August 6:00 UTC, Fig. 3g), cloud (BT ≤ 210 K) originating from the Sichuan convective cells are reaching the region south of Kathmandu (~85°E) while some thin clouds (BT ≤ 250 K) partially cover the region
~85°E on the flight track. The BT distributions derived from the CNTL simulation (Figs. 3b, d, f, h) show that the life cycle of deep convective clouds that occurred over the Sichuan Basin during 7−8 August is correctly reproduced by the simulation. While the CNTL simulation overestimates the extension of convective cloud at 12:00 UTC on 7 August (Fig. 3a, b), it realistically captures the area covered by the deep convective clouds (BT ≤ 210 K) at 18:00 UTC (Figs. 3c, d), and the thinner clouds corresponding to the convection outflow which are horizontally-stretched to the west between 00:00 UTC and
06:00 UTC on 8 August (Figs. 3e−h).

## 3.2 Chemical composition

The simulated chemical data were interpolated to the height, latitude, and longitude of the aircraft rounded to the nearest 40 s time step of the model for a direct comparison. The comparison between observed and simulated CO (Figure 4) shows that CNTL succeeds to reproduce the CO variations along the tracks of all flights. In particular the simulation reproduces very
well the anti-correlation between the CO concentration and the altitude as expected from a tropospheric tracer.

Flight #5 (Fig. 4a) consists in a step by step ascent from 16 to 20 km with the observed-CO concentrations (green circles) gradually decreasing from 90 to the stratospheric background of 10 to 20 ppb. The CO concentration from the CNTL simulation follows the observed one deceasing from 150 to ~20 ppb. During the fast ascent and descent part of the flights, at altitudes below 15 km, the simulated concentrations exceed the observed ones. Flight #6 flew mostly at about 17 km altitude
with the observed-CO concentrations about 60 ppb. The CNTL simulation generally overestimates this low CO concentration by ~20 ppbv. At around 34000 s the aircraft goes into a dive at 15 km and the CO concentration rapidly increases up to about 140 ppb in both measurement and simulation. Before and after the dive, the observations display two peaks of 120 ppbv CO, which are not well reproduced by the simulation, where increases are closer in time to the dive and last longer. Flight #7 (Fig. 4c) is characterized by excursions within the UTLS between 15 and 20 km. Both measurements and simulation show the
clear anti-correlation between altitude and CO concentration, which ranges from 10 to 130 ppbv. Finally, during flight #8 (Fig. 4d), the aircraft remains at ~17 km (86 hPa) during the 35000−40000 s period near Kathmandu foothills. During this period, the measured CO concentrations remain within a 30−50 ppb background except for three 90 ppb peaks at the



beginning. During the same time, CO from the CNTL simulation gradually decreases from 90 to 55 ppb. As for flight #6 it appears that the model is not able to reproduce the short CO peaks but instead produces longer and smoother increases. This

is probably linked to a too coarse model grid spacing not adapted to capture fine plumes. At the end of the flight, the aircraft ascends to 19.4 km and the simulation overestimates the concentrations by up to 30 ppb. About 40 ppb overestimation is also clear for the tropospheric profiles during take-off and landing.

Contrarily to CO, as a stratospheric tracer, for the three StratoClim flights (Figure 5), $O_3$ measured by the FOZAN instrument (red circles) is highly correlated to flight altitude (blue line). It is particularly clear with flight #5 where $O_3$ is

increasing stepwise such as the aircraft altitude until it reaches 1055 ppbv at 20.1 km. The CNTL simulation (black lines) is able to capture accurately those observed variations with a good agreement with the measured-$O_3$ except in the troposphere below about 12 km where it is overestimating the low $O_3$ concentrations, which are decreasing down to 0 ppbv. During flight #7, the $O_3$ variations (80−600 ppbv) related to the altitude excursions of the aircraft are very well captured by the model but the modelled-$O_3$ is high biased by 50 to 80 ppbv during this flight. It is noteworthy that $O_3$ concentrations measured around

Kathmandu (about 22500 s, 19.3 km) are one third lower during flight #7 than during flight #5. This is probably linked to the transport of relatively fresh BL air masses from the Tibetan Plateau (Bucci et al., 2020). Observed-CO concentrations corroborate this hypothesis with lower CO during flight #5 (Fig. 4a) than during flight #7 (Fig. 4c) around Kathmandu. During flight #8 (Fig. 5c), near constant $O_3$ concentrations are measured at the near constant flight altitudes, i.e. ~120 ppbv at 18 km and ~400 ppbv at 19 km. Correlatively, $O_3$ from the CNTL simulation ranges from 170 to 400 ppbv overestimating the

concentrations by ~100 ppbv during 38000−40000 s when the aircraft is flying at 18 km over southern foothill of Kathmandu (~200 ppbv instead of ~100 ppbv).

During the StratoClim period, we benefit from two commercial flights from the IAGOS program across the south-western part of the AMA (Fig. 6) at the near constant altitude of ~11.5 km at bottom of the AMA. Both measurements and CNTL simulation display clear and coincident CO enhancements within the AMA range. At the north-western edge of the AMA

(30°N, 60°E; red arrows), the aircraft penetrates the tropical air masses from the extratropical UTLS and CO rapidly increases from 80 to 110 ppb. At the south-western edge of the AMA (~15°N, 75°E; black arrows) CO gradually decreases from AMA high concentrations to lower tropical UT concentrations. The model correctly captures both rapid extratropical UTLS to AMA and slow AMA to tropical UT transitions along the flight tracks at the bottom of the AMA (~11.5 km). The model overestimates the CO concentrations by 5 to 20 ppb within the AMA range.

IAGOS measured-$O_3$ (Fig. 6) slowly increases from 46 ppbv in the tropical UT south of the AMA to about 90 ppbv in the extratropical UTLS north of the AMA. The CNTL simulation displays the same increasing trend with nonetheless an overestimation of ~20 ppbv $O_3$ in the AMA and tropical UT and ~50 ppbv in the extra-tropical UTLS which is consistent with the high biases relative to FOZAN $O_3$ from StratoClim Geophysica flights.

In summary, comparisons with 10.8 microns BT images from geostationary satellites have shown that the location and

intensity of deep convective clouds are very well reproduced by the CNTL simulation suggesting that the model is able to reproduce the convective uplift of BL pollutants into the UT. Comparisons with StratoClim and IAGOS in situ observations





have highlighted the ability of the model to reproduce largely CO and $O_3$ UTLS variations from 10 to 21 km. It is noteworthy that the CNTL simulation tend to overestimate CO by up to 20 ppbv in the TTL (Tropical Tropopause Layer) especially around Bangladesh (flight #6) at ~17 km and at the southern foothills of Kathmandu (flight #8) at ~18 km and 40 ppbv in the troposphere at take-off and landing. $O_3$ is overestimated by ~20 ppbv in the AMA an up to 100 ppbv south of Kathmandu at ~18 km (flights #7 and #8). Overestimation also reaches 45 ppbv in the extratropical UTLS at ~11.5 km relative to IAGOS data.

## 4 Convective uplift of Sichuan pollution to the AMA

We have shown that Meso-NH was able to reproduce the strong deep convective event that took place over Sichuan on August 7 (Fig. 3). In this section we document the impact of this isolated and unusual convective event on the composition of the AMA and the transport pathway of pollutants from the BL of Sichuan to the Indian UTLS.

We have selected five times of analysis: 1) the time of deep convection start, 06:00 UTC, 2) the time of deep convection development, 12:00 UTC, 3) the time of matured deep convection, 18:00 UTC on 7 August, 4) the time of dissipating deep convection, 00:00 UTC, and 5) the aircraft measurement time, 06:00 UTC on 8 August. We define a 'CO patch' as a region with a CO concentration larger than the average concentration of the entire AMA at 14.8 km at 06:00 UTC on 7 August (160 ppbv) using CNTL simulation results. The altitude setup of 14.8 km results from a trade-off between having influence of strong convective updraughts and being in the lower entry region of the AMA. The horizontal distributions of 10.8 μm BT, CO and horizontal winds at 14.8 km from the CNTL experiment are displayed in Figure 7. To understand the contribution of the Sichuan emissions to the AMA composition, a sensitivity experiment, named 'SIC06' was conducted (Table 2). In SIC06 the emissions of the Sichuan Basin (26−33°N, 101−109°E, red box in Fig. 2) are set to zero between 18:00 UTC on 6 August and 00:00 UTC on 8 August, period which encompasses the whole development of the 7 August convective event. All the other environmental conditions are identical to the CNTL run.

The contributions of the uplifted Sichuan emissions to the CO concentration at 14.8 km corresponding to the difference with the CNTL simulation (= [CNTL minus SIC06] over [CNTL]) are displayed in Figure 7 (third column). During the whole period, the CO patch is highly coincident with the deep convective system that develops over the Sichuan Basin (delimited by a box). In order to document the vertical transport of pollution and its relationship with convective clouds, Figure 8 displays the longitude-altitude cross sections of simulated CO concentrations and winds together with cloud contours corresponding to the locations of the CO patches of Fig. 7 (for the location, see black solid lines in Fig. 7).

The AMA covers a large area (20–35°N, 60–120°E) characterized by the anticyclonic circulation and the enhanced CO concentrations (middle panels of Fig. 7) from the embryo of convection at 06:00 UTC on 7 August are dissipated at 06:00 UTC on 8 August. At the mature stage of convection (18:00 UTC on 7 August), CO concentrations reach maximum values of 195 ppbv within the convective core.





At the start of the deep convective event (06:00 UTC on 7 August) the embryo of the CO patch (enhancement of ~5 ppbv) is coincident with low BT values ($\leq 210$ K, Fig. 7a), enhanced CO concentrations ($\geq 140$ ppbv) and the northerly winds ($\leq 15$

m s$^{-1}$, Fig. 7b) from the eastern edge of the AMA over the Sichuan Basin (box, Fig. 7a, b). The vertical cross section across the CO patch embryo (Fig. 8a, latitude 29.5°N) highlights the accumulation of a large amount of pollution in the Sichuan Basin at the foothills of the Tibetan plateau (around 104°E). A plume of enhanced CO concentrations is the most visible between 6 and 9 km below the convective cloud which clearly overshoots the 380 K isentrope above the plume. During the development of convection (12:00−18:00 UTC, Figs. 7d−i) low BT ($\leq 210$ K) and enhanced CO concentrations ($\geq 160$ ppbv)

with a large Sichuan contribution of 5−25 ppbv (or 5−15 %) spread over most of the Sichuan. At 12:00 UTC (Fig. 8b), the convective clouds cross the 380 K isentrope over a larger area and CO is uplifted by the intense convective updraughts from the mountain foothills of Sichuan to the TTL and the 380 K isentrope. At 18:00 UTC (Fig. 8c), the horizontal CO divergence is observed in the UT. The CO plume which has been transported to the south-west before 18:00 UTC is caught by north-easterlies to be further advected westward (Figs. 7k, n). At 00:00 UTC on 8 August, the plume has come out of the

convective region with its eastern part still embedded in a high altitude (11−16 km) cloud and its highest part (14−16 km) stretched westward by the strong easterlies ($\geq 35$ m s$^{-1}$) (Fig. 8d). At the time of Flight #7, the CO plume that extends over the 15−17 km altitude range (Fig. 8e) has been diluted with concentrations lower than 170 ppbv (Fig. 7n) and Sichuan contributions to total CO is lower than 10 % (Fig. 7o). In the region south of Kathmandu sampled by the Geophysica aircraft, the Sichuan contribution is reduced to about 2 % (~5 ppbv) at 14.8 km.

The CO concentrations along Flight #7 resulting from SIC06 simulation are displayed by a dashed line in Figure 4c. Depending on the altitude the impact of Sichuan emissions range from 3 to 10 ppb. The most interesting differences between CNTL and SIC06 occurs between 19200 and 20000 s with the aircraft first descending from 17 to 14 km where it remains about 4000 s before ascending to 18.5 km. During the descent and the ascent, between 15 and 16 km, the measured CO concentrations are larger by ~10 ppbv than at 14 km resulting in two small CO peaks. The CNTL simulation reproduces these

two peaks fairly well while they are absent from the SIC06 simulation (dotted line, Fig. 4c). The differences between the two simulations (~5 ppbv) is about half of the measured one. Nevertheless, the coincidence of measured and simulated peaks allows us to attribute to Sichuan emissions an increase of CO about 3−12 ppbv in the region of Kathmandu between 15 and 16 km on 8 August.

To further determine the impact of Sichuan emissions uplifted by the 7 August convective event we have computed the

relative difference between CNTL and SIC06 simulations averaged over the AMA domain (20−35°N, 60−120°E). The evolution of the difference is displayed in Fig. 9 with the blue bars on top indicating the surface covered by deep convective clouds. The impact in the lowermost layer (0.1−5 km) starts as soon as the emissions are turned off in the SIC06 simulation and increases steeply until the convection has developed on 7 August 12:00 UTC. Afterwards, the increase slows down in the lower layer because convection transports pollution in the upper layers where the impact increases. The impact remains very

limited in the 5−10 km layer and is more important in the 10−15 and 15−20 km layers because convection detrainment occurs above 10 km. The impact decreases in the 0.1−5 km layer as soon as the emissions are back to normal in SIC06 (8



August 00:00 UTC) but remains steady (~0.5 %) until 9 August 12:00UTC in the two uppermost layers because the CO convective plume is trapped within the AMA. The sudden release of BL emission just before the convective activities in the Sichuan Basin attributes the rapid increase of ~0.5 % in CO into the AMA.

The CO plume identified in Figs. 7 and 8 traces pollution emissions from the Sichuan BL into the AMA. In order to have an insight about the impact of Sichuan pollution on the ATAL formation we looked at the behaviour of simulated primary aerosols emitted by the same sources as CO. The POA (Primary Organic Aerosols) and BC (Black Carbon) distributions at 14.8 km for both size modes are displayed in Figure 10 for August 7 at 18:00 UTC. As expected the POA coarse particles mode #2 (accumulation mode; for details about size distribution, see section 2.3) has a much larger contribution to the mass
concentration with maxima of ~5.5 µg m$^{-3}$ over the Sichuan Basin than the small particles mode #1 (Aitken mode; maxima of ~0.5 µg m$^{-3}$). BC peak mass concentrations are about six times lower than POA with the same repartition between mode #1 and mode #2.

The longitude-altitude cross sections of the particles mass concentrations (along 28.3°N) displayed in Fig. 11 highlight their convective uplift over the Sichuan Basin. As in Fig. 10, POA are much larger than BC but the main features are
identical for both types of particles. Particles of Aitken mode (mode #1) follow the same transport pathway than CO (Fig. 8c) with uplift in the convective updrafts up to the 380 K isentropic level around 104°E. Relative to CO their concentrations is largely reduced with only about one third remaining above 12 km following scavenging within the convective clouds. Large particles of accumulation mode (mode #2) enhancement is not even detectable within the convective pipe above 12 km. Indeed large particles are more efficiently scavenged by cloud and rain droplets than small particles. The initial mean aerosol
radius of mode #1 and #2 are 0.036 µm and 0.385 µm, respectively, which is too small to be effectively scavenged by convective precipitation below and in the cloud (Slinn et al., 1983). This result thus implies that aerosol sizes in both modes within the polluted plume are increased during the uplifting within the cloud by gas-particles conversion, condensation of water in the aerosol and coagulation (Andronache, 2003; Tost et al., 2007; Berthet et al., 2010; Tulet et al., 2010). The topography (top altitudes ~4 km) around the Sichuan Basin triggered the continuous deep convective events which ventilated
the large amounts of pollutants, i.e. CO, POA, and BC, stored at the mountain foothills into the AMA.

## 5 Impact of regional sources on UTLS composition

The previous section has highlighted the impact of a single strong convective event from a highly polluted region on the composition of the AMA. It can raise the CO concentration by up to 18 % in the convective area and by up to 10 % over a large region. The impact over the whole AMA reaches a maximum of about half a percent and it is largely reduced after a
couple of days. In order to determine the impact of emissions from the neighbouring regions to the composition of the AMA during the StratoClim campaign, three additional experiments (Table 2) have been conducted with emissions turned off over Sichuan (SIC01, 26−33°N, 101−109°E, red box in Fig. 2) over China (CHN01: 20−40°N, 100−122°E, black box) and over India (IND01, 10−35°N, 70−95°E, green box) at 00:00 UTC on 1 August until 00:00 UTC on 8 August. Figure 12 displays





10-days averaged horizontal distributions of CO at 14.8 km from the CNTL simulation and the differences between CNTL

and sensitivity experiments SIC01, CHN01, and IND01. The CO concentrations from SIC01, CHN01 and IND01 along the Geophysica flight tracks are displayed in orange, red and blue in Fig. 4. For the four analysed flights, the local Indian contribution is generally larger than the Sichuan and Chinese remote contributions. This is in agreement with Bucci et al. (2020) who quantified the convective contribution of different source regions to the CO concentration along the StratoClim flight tracks using a Lagrangian dispersion model. The Indian larger impact is especially clear at take-off and landing at

altitudes where Chinese air masses are not efficiently transported by the UTLS easterlies.

Flight #5 (Fig. 4a) took place over the Nepalese region and sampled the 16−20 km altitude range before the strong convective events in China. Therefore, as documented in Bucci et al. (2020), there is no Chinese contribution during this flight. The local (Indian subcontinent) BL sources contribution since the start of the simulation (1 August 00:00 UTC) to the CO sampled up to 18 km is 10−15 ppbv ($\geq$ 10 %). According to the model, recent convection does not provide local fresh

pollution above this altitude.

During flight #6 the aircraft flew back and forth from Kathmandu to the Bay of Bengal coast over Bangladesh (Fig. 1) and remained at the constant altitude of ~17 km (Fig. 4). During the whole flight at 17 km the Chinese contribution is larger than the Indian one, which remains weak or null. According to Bucci et al. (2020), the two largest CO peaks at ~32500 and ~35500 s are resulting from Sichuan BL air masses convectively uplifted 2 days before. As mentioned in section 3.2, the

Meso-NH model is not able to capture these 2 peaks but rather smooth CO enhancements. The CHN01 sensitivity simulation highlights the Chinese origin of enhanced CO between 32000 and 37000 s similarly to Bucci et al. (2020) in their Fig. 7. The SIC01 simulation suggests more specifically that the origin of enhanced CO is not Sichuan contrarily to Bucci et al. (2020). The model missed the convective event responsible for these particular peaks and is therefore not able to reproduce them. Bucci et al. (2020) pointed to air masses from the Southeast Asian peninsula and South China recently uplifted to lower

levels (13 km) for the highest CO peak corresponding to the aircraft dive at 15 km. Our sensitivity simulations confirm the Chinese (excluding Sichuan) contribution and the almost zero Indian contribution to this peak. The differences could be partly induced by the differences in emission data (i.e. MIX vs MACCity) and the slightly different cloud location in observation vs model.

Flight #7 has already been discussed about the impact of the 7 August strong convective event in Sichuan. In agreement

with Bucci et al. (2020) and their Fig. 10, we find an important Chinese contribution all along this flight. As the CO from CHN01 and SIC01 are almost similar, this Chinese contribution mostly originates from Sichuan.

Flight #8 of 10 August was intended to sample the outflow of an intense convective system that had developed over the Ganges valley. During the first leg of the flight, the aircraft flew in the inner core of the AMA around the altitude of the tropopause (85−90 hPa) sampling old air that had been convectively injected into the UTLS mostly more than 10 days ago

(Bucci et al., 2020). This means that the Meso-NH sensitivity simulations cannot fully determine the source apportionment for CO. Nevertheless, between 35000 and 38000 s about 10 ppbv CO can be attributed to BL air masses from both the Indian and Chinese domain uplifted after first August 00:00 UTC. The Chinese and Indian contributions are also detected for this



period of the flight in Bucci et al. (2020). After 38200 s and during the stratospheric part of the flight (after 40200 s) and before landing, the Chinese and Indian contributions from less than 10 days before are negligible. The analysis of the CO
from our Meso-NH simulations along the StratoClim flights in light of the study of Bucci et al. (2020) have allowed us to further validate the model. It is able to correctly reproduce the convective uplift and transport pathways of pollution from the main Asian regions to the AMA even if some convective events resulting in UTLS CO peaks were missed.

Interestingly between 42500 and 43000 s in Flight #8 and between 22600 and 22800 s in Flight #7 CO from the CHN01 simulation is lower and in better agreement with observed CO −which remains close to stratospheric background− than CO
from the CNTL simulation. This implies that the contribution of Chinese pollution to the lower stratosphere is slightly overestimated by the model and that the latest contribution of BL pollution to this altitude probably dates back several weeks.

Looking at the UT CO distributions from our simulations allow us to have a broader view of the average impact of the source regions. CO distribution from the CNTL run (Fig. 12a) is characterized by two large regions of high CO ($\geq$ 120 ppbv) the first one at the foot of the Himalayas encompasses large parts of Pakistan, northern India, Nepal and Bangladesh
(20−30°N, 70−95°E) and the second one large parts of central China (100−120°E, 20−30°N). The CO distribution of CNTL minus CHN01 (Fig. 12b) highlights that about 20 % (~25 ppbv) of CO over central China comes directly from China. Otherwise, the uplifted Chinese emissions follow the easterlies and affect mostly the southern part of the AMA from southern China to the Arabian Sea with a 12−15 ppbv contributions. The emissions from the Sichuan Basin alone account for about 7 % (~10 ppbv) of CO over central China and also follow the easterlies but north of the main Chinese plume (Fig. 12c). The
10-days average contribution of Chinese emission to the whole AMA is 6.9 % (8.0 ppbv) in the 10−15 km layer and reaches 3.3 % (1.9 ppbv) in the 15−20 km layer. These values are smaller than the contribution of East Asia (10 ppbv) computed by Barret et al. (2016) using simulations from the GEOS-Chem model for the month of August 2009. Note the difference time scale used in studies (10 days vs 1 month) since AMA composition varies with a typical period of two weeks (cf. Bucci et al., 2019) and even a month and a season.

Indian emissions (Fig. 12d) are impacting the whole AMA from the Arabian sea to the Pacific but most notably its central region west and south of the Himalayas (Bangladesh, northern India, Pakistan, Afghanistan) with a contribution of 20−25 % (25−30 ppbv) to the CO concentration. Averaged over the 1−10 August, the Indian domain contributed to 10.9 % and 6.2 % (12.8 and 3.7 ppbv) CO in the layers from 10 to 15 km and from 15 to 20 km, respectively. Barret et al. (2016) found a larger contribution of 24 ppbv for their South Asian domain. As in Barret et al. (2016) we find that the Indian (South Asian)
contribution is about 1.6 times of the Chinese (East Asian) contribution to CO in the AMA.

## 6 Summary

This paper focuses on the emission sources and pathways of pollution from the BL to the AMA during the StratoClim campaign period. To that purpose, we have performed cloud-chemistry simulations with the Meso-NH model with a 15 km horizontal and 100 to 450 m vertical resolutions. To validate the simulated clouds, we have used a combination of the IR



window data from the MSG/SEVIRI and Himawari/AHI geostationary instruments. The CO and $O_3$ distributions have been compared with in situ airborne observations from the StratoClim campaign in the TTL region and from the IAGOS program in the UT. The comparison of IR window BT demonstrates the ability of the model to reproduce the deep convective clouds over the whole Asian region but more specifically the intense events that took place over the Sichuan Basin on 7 August 2017. Comparisons with StratoClim and IAGOS measurements show that Meso-NH reproduces correctly the variations of

both CO and $O_3$ in the Asian UTLS from 11 to 21 km. Nevertheless, CO concentrations are regularly overestimated by 20 ppbv in the TTL and even 40 ppbv in the troposphere. This gap could be influenced by the selected emission and initial atmospheric chemistry data (i.e. MOZART) in this study. $O_3$ is also overestimated by ~20 ppbv in the troposphere and TTL and up to 100 ppbv in the extratropical UTLS or in the AMA close to the 380 K isentropic level (17.8 km).

    During the StratoClim period, exceptionally strong convection occurred in the Sichuan Basin at the Tibetan plateau

foothills (east of 90°E) with clouds reaching the tropopause over large areas and overshooting the local tropopause. The SIC06 simulation was dedicated to the characterisation of the transport pathway of BL pollution from the Sichuan to the UTLS and on the impact of the 7 August convective event on the composition of the AMA. The results show that BL CO convectively uplifted up to 18 km (380 K isentropic level at 17.8 km) contributes to more than 15 % (~25 ppbv) to the CO concentration within the convective core where it reaches 195 ppbv. The CO plume is further transported westward by the

strong north-easterlies and easterlies on the edge of the AMA and reaches the region south of Kathmandu at the time of StratoClim Flight #7. The aircraft mostly flew above the most impacted altitudes range (15−17 km) and the plume had been largely diluted when it reached the flight track. Nevertheless, the difference between the CNTL and the SIC06 simulations demonstrates that two peaks corresponding to 10 ppbv CO anomalies detected during flight #7 between 14 and 16 km most likely result from the Sichuan pollution plume. Over the whole AMA, the impact of the 7 August Sichuan convective event

contributed to ~0.5 % in the layer between 10 and 20 km during 2 days. The Meso-NH simulation also documents the uplift of carbonaceous primary particles (BC and POA) to the tropopause over Sichuan and their further transport westward. Interestingly, particles of accumulation mode (initial mean radius of 0.385 μm; standard deviation of 1.29) are increased during the uplifting within the cloud and are more scavenged within the convective clouds but their contribution remains much more important to the mass concentration in the AMA.

Dedicated sensitivity simulations showed that the Indian (IND01 run) and Chinese (CHN01 run) domains mostly impact the AMA locally with contributions of 25 to 30 ppbv. The Chinese contribution is advected westward at the southern edge of the AMA by the easterlies while the Indian one remains within the core region of the AMA. The Chinese and Indian contributions to the CO simulated along the StratoClim flight tracks have been compared to results from Lagrangian dispersion modelling (Bucci et al., 2020). Despite some important methodological differences, we found a good general

agreement with nonetheless some CO peaks originating from recent convection (from Southeast Asia and Sichuan) not captured by Meso-NH. For the whole AMA averaged over ten days, the Indian contribution (11 %) is about 1.6 times larger than the Chinese one (7 %). The Chinese contribution is mostly from Southern China where monsoon convection is the largest (Bucci et al., 2020). Nevertheless, the SIC01 simulation highlights that Sichuan represented an important part (35 %)

of the Chinese contribution as a result of the strong convective events that occurred in this region during the StratoClim

campaign. Comparisons with Barret et al. (2016) have shown that smaller contributions are derived during the period of 1−10 of August 2017 compared to the month of August 2009. Further studies will focus on the formation and transport pathways of particles which form the ATAL and on simulations at higher horizontal resolution (< 5 km) which will resolve deep convection.

*Data availability.* Stratoclim data will be freely available at the https://halo-db.pa.op.dlr.de/mission/101 database from the

end of June 2020, in the meanwhile they will be available upon request to the authors. Meso-NH output data are available from Eric. L. Flochmoën upon request. The satellite data and the emission data are freely provided by the AERIS data centres (https://en/aeris-data.fr/).

*Author contribution.* KOL and BB designed the numerical simulation, manuscript and analyses. ELF, BB, KOL, and PT performed numerical simulations. ML contributed to the Meso-NH configuration for the chemistry part. MVH, CK, FR, and

AU provided the aircraft instrumental data and BL and SB provided satellite data. KOL prepared the manuscript with contributions from all co-authors.

*Competing interests.* The authors declare that they have no conflict of interest.

*Acknowledgement.* Authors thank Céline Marie (Laboratoire d'Aérologie) for her insightful suggestions. This work was

supported by French ANR TTL-Xing ANR-17-CE01-0015 project and the StratoClim project by the European Community's Seventh Framework Programme (FP7/2007−2013) under grant agreement no. 603557. We also thank CEFIPRA support through grant 5607-1. IAGOS data were created with the support from the European Commission, national agencies in Germany (BMBF), France (MESR), the UK (NERC), and the IAGOS member institutions (http://www.iagos.org/partners). The participating airlines (Lufthansa, Air France, Austrian, China Airlines, Iberia, Cathay Pacific, Air Namibia, and Sabena)

supported IAGOS by carrying the measurement equipment free of charge since 1994. The data are available at http://www.iagos.fr thanks to additional support from AERIS. Meteorological analysis data are provided by the European Centre for Medium-Range Weather Forecasts.

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





**Table 1.** Duration and variables of StratoClim flights. For the flight track, see Figure 1.

| Exp. | #5 | #6 | #7 | #8 |
|---|---|---|---|---|
| Duration | 03:00−07:25UTC 4 August | 07:30−11:30 UTC 6 August | 04:30−06:50 UTC 8 August | 08:40−12:30 UTC 10 August |
| Variables | CO, $O_3$ | CO, $O_3$ | CO | CO, $O_3$ |

**Table 2.** List of sensitivity experiment and duration and area of emission modifications. For the area of modification, see Figure 2.

| Exp. | SIC06 | SIC01 | CHN01 | IND01 |
|---|---|---|---|---|
| Duration | From 18:00 UTC 6 August to 00:00 UTC 8 August | From 00:00 UTC 1 August to 00:00 UTC 8 August | From 00:00 UTC 1 August to 00:00 UTC 8 August | From 00:00 UTC 1 August to 00:00 UTC 8 August |
| Area | Sichuan Basin | Sichuan Basin | Centre China | India |


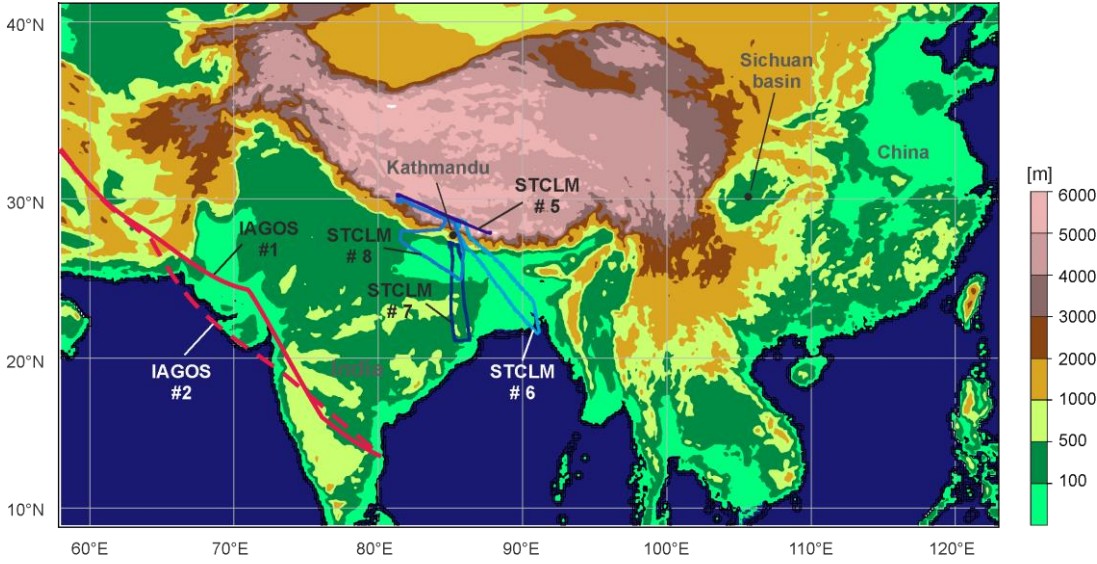

**Figure 1.** Topography and domain considered in the Meso-NH numerical simulation with a resolution of 15 km. The trajectory of the
Geophysica flights #5, #6, #7, and #8 during the StratoClim campaigns (marked with 'STCLM') around and south of Kathmandu are
shown by the bluish solid lines, while two IAGOS flight tracks (to/from Madras in India) are indicted by the red solid and dashed line.





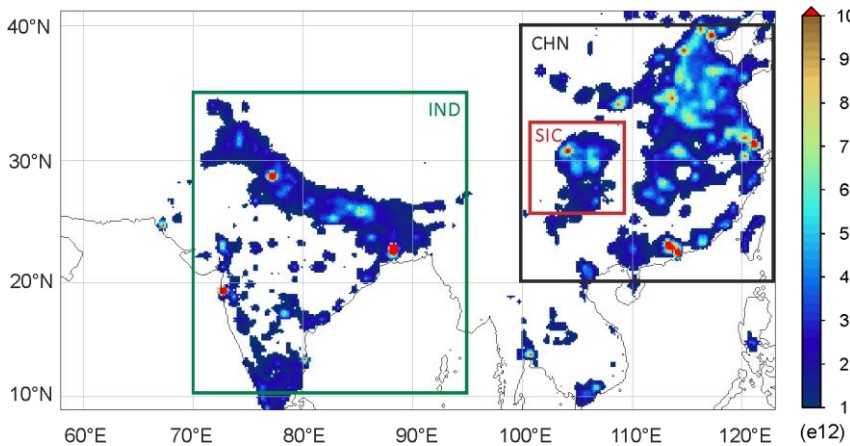

**Figure 2.** The emission map of carbon monoxide used for CNTL. Inner boxes indicate the domain of no emission for sensitivity experiments (see also Table 2) of SIC06 and SIC01 (101−109°E, 26−33°N, red line), CHN01 (100−122°E, 20−40°N, black line), and IND01 (70−95°E, 10−35°N, green line) simulations.








**Figure 3. Infrared** BT composite images using SEVIRI/MSG and Himawari (left) and Meso-NH (right) at (a), (b) 12:00 UTC, (c), (d) 18:00 UTC on 7 August, (e)−(f) 00:00 UTC, and (g)−(h) 06:00 UTC on 8 August. The area of interest is marked by yellow box.

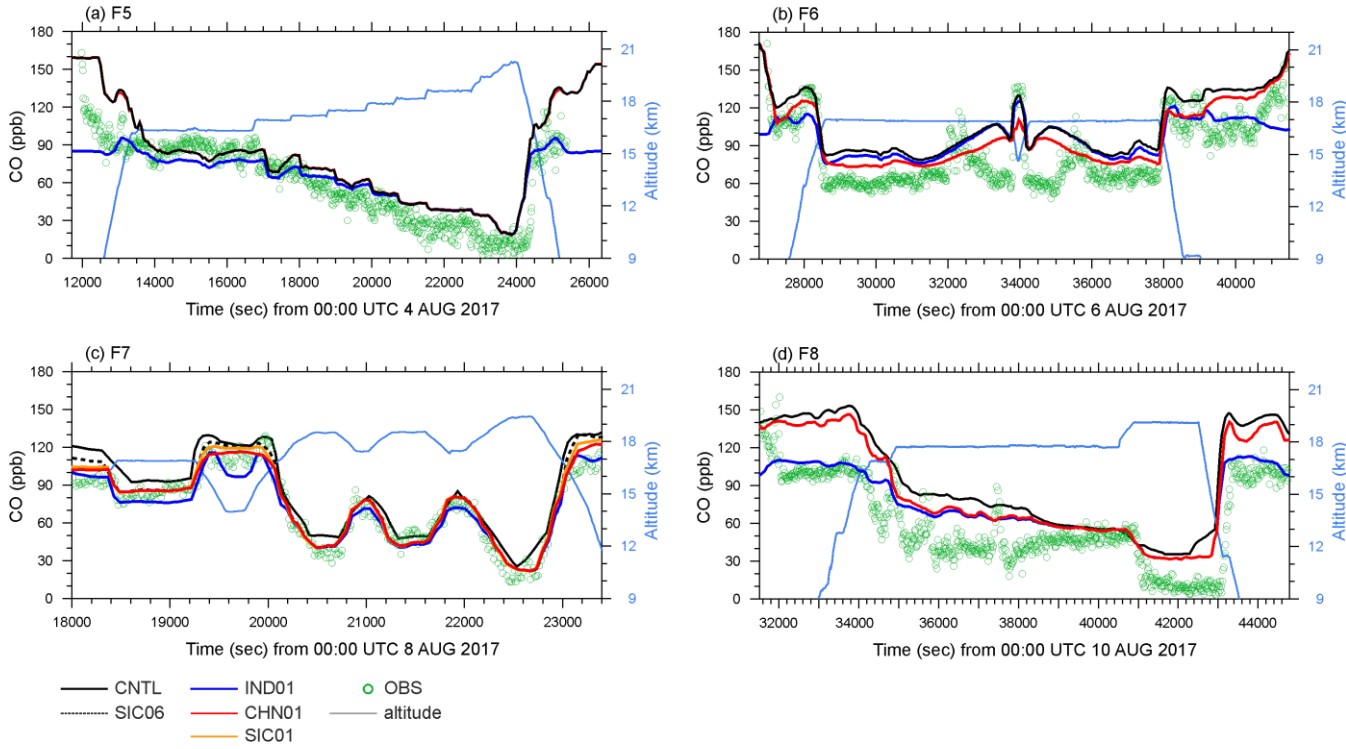


**Figure 4.** Comparison of carbon monoxide (ppbv) of AMICA-measured (green circles) and Meso-NH-derived (black lines) along the StratoClim flight tracks (a) #5 on 4 August, (b) #6 on 6 August, (c) #7 on 8 August, and (d) #8 on 10 August 2017 (for the location, see Figure 1). The simulated CO by SIC01, CHN01, and IND01 along the flight tracks are displayed by orange, red, and blue solid lines, respectively. Flight altitudes are displayed by light blue lines. In (c), the SIC06-produced CO along the flight track is displayed by a dotted line.




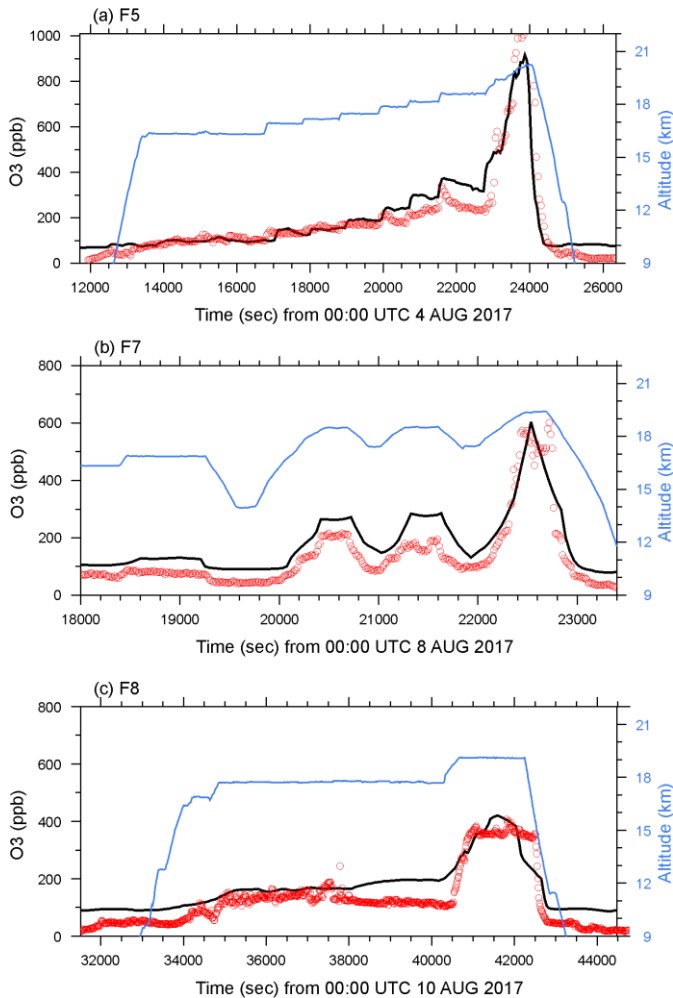

**Figure 5.** Comparison of ozone (ppb) of measured (red circles) and Meso-NH derived (black lines) along the StratoClim flight tracks (a) #5 on 4 August, (b) #7 on 8 August, and (c) #8 on 10 August 2017 (for the location, see Figure 1). Flight altitudes are displayed by light blue lines.

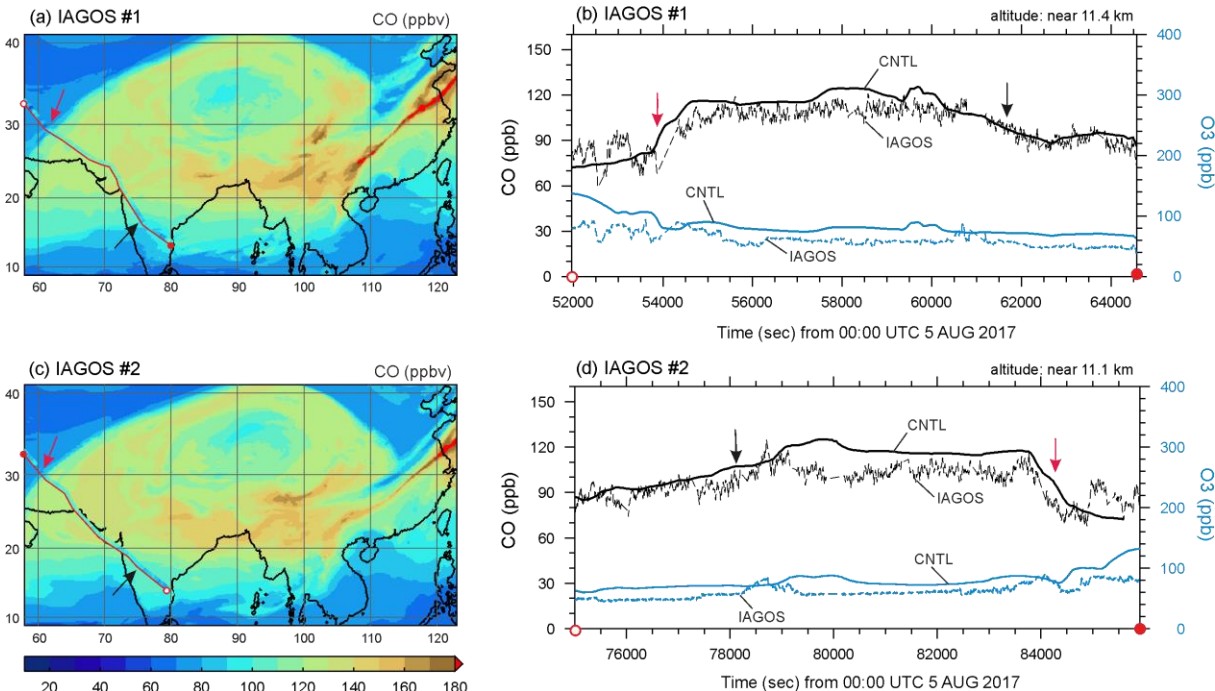

**Figure 6.** IAGOS-measured (dashed lines) and Meso-NH-derived (solid lines) carbon monoxide (black lines) and $O_3$ (blue lines) along IAGOS flight tracks on 5 August 2017. In (a) and (c), Meso-NH-derived CO at the altitude of 11.1 km are displayed by shaded areas, while the IAGOS-measured CO are displayed by coloured circles along the track (red lines). In (a)–(d), the starting (ending) point of each flight within the domain is marked by open (closed) red circle, while the location of the steep (gradual) change of carbon monoxide is marked by red (black) arrows.




**Figure 7.** Horizontal map of BT 10.8µm (K, left), and carbon monoxide (ppbv, middle) and CO contributed by Sichuan emission (%, right) at the altitude of 14.8 km at (a)−(c) 06:00 UTC, (d)−(f) 12:00 UTC, (g)−(i) 18:00 UTC on 7 August, (j)−(l) 00:00 UTC, and (m)−(o) 06:00 UTC on 8 August 2017. The area of interest is marked by black or yellow (for visibility) box. In right panels, the location of longitude-altitude cross section used in Figure 8 is marked by a black solid line.

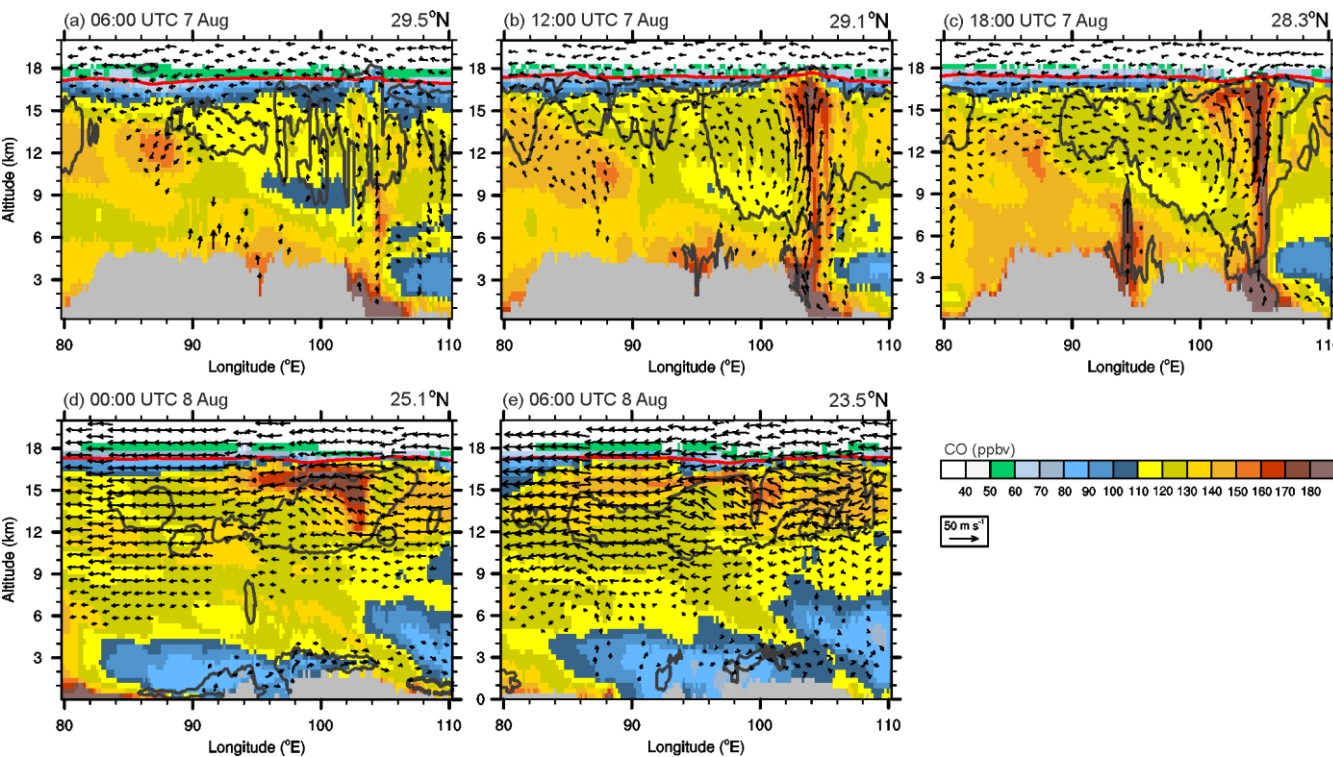

**Figure 8.** Vertical cross sections of CO (ppbv) from 0.1 km to 20 km altitude at (a) 06:00 UTC, (b) 12:00 UTC, (c) 18:00 UTC on 7 August, (d) 00:00 UTC, and (e) 06:00 UTC on 8 August 2017. The isentropic altitudes of 380 K is depicted by the red lines. The cloud boundary (mixing ratio of ice content of 10 mg kg$^{-1}$) is contoured by the black line while wind stronger than 5 m s$^{-1}$ is shown by vectors. In each panel, grey areas indicate the topography.





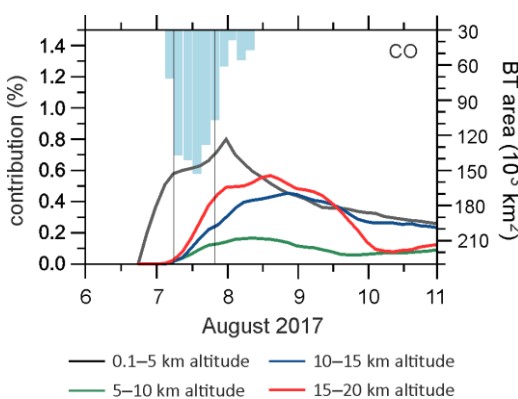


**Figure 9.** Temporal evolution of contribution of Sichuan emission to carbon monoxide concentration of entire AMA region (60−120°E, 20−35°N) every 5 km from 0.1 km to 20 km altitude from 6 to 9 August 2017. The area of low brightness temperature (210 K) in the Sichuan Basin (101−109°E, 26−33°N) are displayed by blue bar.


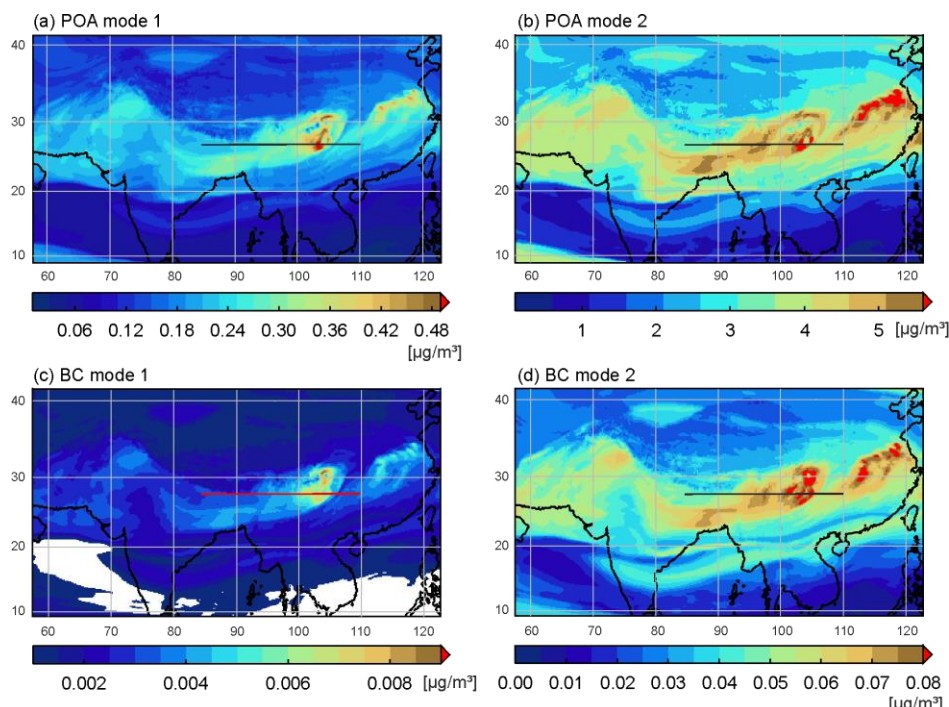

**Figure 10.** Horizontal map of (a) POA of mode #1, (b) POA of mode #2, (c) BC of mode #1, and (d) BC of mode #2 at the altitude of 14.8
km at 18:00 UTC on 7 August. The location of vertical cross sections used in Figure 11 is marked by a black (or red for visibility) solid
line in each panels.





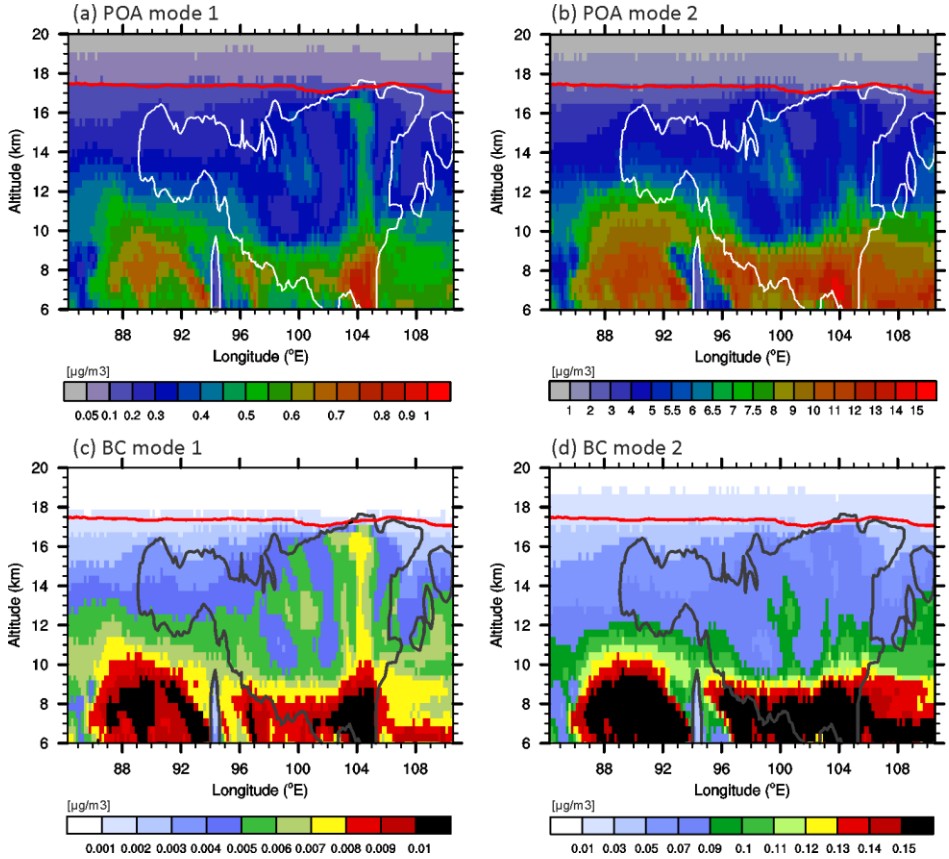

**Figure 11.** Vertical cross sections of (a) POA of mode #1, (b) POA of mode #2, (c) BC of mode #1, and (d) BC of mode #2 from 6 km to 20 km altitude along the x-axis of the area of interest at 18:00 UTC on 7 August. The isentropic altitudes of 380 K is depicted by the red lines. The cloud boundary (mixing ratio of ice content of 10 mg kg$^{-1}$) is contoured by the black or white line (for visibility).



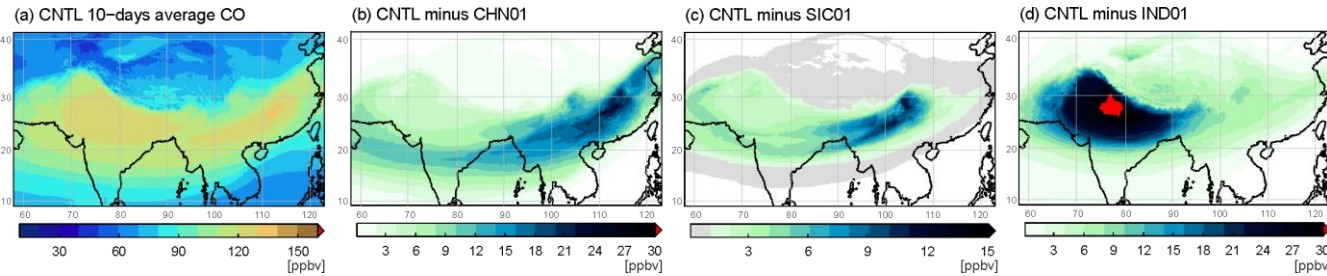


**Figure 12.** 10-days averaged chemical components of CO (ppbv) from 1 to 10 August 2017 produced by (a) CNTL, (b) CNTL minus CHN01, (c) CNTL minus SIC01, and (d) CNTL minus IND01.