# Peer review of "Convective uplift of pollution from the Sichuan basin into the Asian monsoon anticyclone during the StratoClim aircraft campaign"

_Atmospheric Chemistry and Physics, 2020_

## Referee Comment (RC1) · Anonymous Referee #1 · 17 Jul 2020

This manuscript investigated the convective uplift of pollution from the Sichuan basin into the Asian summer monsoon anticyclone during the StratoClim aircraft campaign in 2017 by simulations with the Meso-NH cloud-chemistry model. After validation with the BT from satellite data and CO and ozone from airborne observations, the simulations are believed to study the impact of Sichuan convection on the AMA composition. Overall, the manuscript shows some interesting results, particularly the role of Sichuan basin as source region of pollution. Some major issues should be addressed before acceptance for publication in ACP. Major issues: 1. How to verify the simulations of aerosol in the tropopause layer? Because no measurements are used to validate the simulation. There are already some published studies, where the observations

of aerosol signal are used to validate the simulations. 2. Could you validate the CO simulations with MLS data? The simulations are not so good for F6 and F8 as shown in Fig. 4, so more validations are in need. MLS CO data will surely show the enhanced CO plume after the deep convection if the results given in this paper are correct. Minor issues: 1. L43-45: A recent paper by Bian et al. (2020) gives a comprehensive review of the deep convection on the UTLS composition during the ASM, which is recommended to be cited here. Bian, J., et al., 2020: Transport of Asian surface pollutants to the global stratosphere from the Tibetan Plateau region during the Asian summer monsoon, National Science Review, 7, 516-533, doi:10.1093/nsr/nwaa005. 2. L52: Higher tropopause over the ASM region is shown by Bian et al. (2012), which also shows the structure of AMA and therefore is recommended to be cited here. Bian, J., L. L. Pan, L. Paulik, H. Vömel, H. Chen, and D. Lu, 2012: In situ water vapor and ozone measurements in Lhasa and Kunming during the Asian summer monsoon. Geophys. Res. Lett., 39, L19808, doi:10.1029/2012GL052996. 3. L64-68: The different contribution from Indian and China sources to the UTLS is investigated by Yan et al. (2015), which conducts the simulation for one month with WRF-chem model and is recommended to be cited here. Yan, R. and J. Bian, 2015: Tracing the boundary layer sources of carbon monoxide in the Asian summer monsoon anticyclone using WRF-Chem. Adv. Atmos. Sci., 32(7), 943–951, doi: 10.1007/s00376-014-4130-3. 4. L180-190: How are the CCN activation and second activation by entrainment in the convective cloud considered, which is critical to the simulation of aerosol profile as shown by Yu et al. (2018)? Yu, P., K.D. Froyd, R.W. Portmann, O.B. Toon, S. R. Freitas, C. G. Bardeen, C. Brock, T. Fan, R. S. Gao, J. M. Katich, A. Kupc, S. Liu, C. Maloney, D. M. Murphy, K. H. Rosenlof, G. Schill, J. P. Schwarz and C. Williamson (2019), Efficient In-cloud Removal of Aerosols by Deep Convection,Geophys. Res. Lett., 45, 1061-1069, https://doi.org/10.1029/2018GL080544 5. Fig 6 and Fig 7: CO data from MLS is suggested to compare with the model simulation. 6. Fig. 10: CALIPSO data for aerosol is suggested to compare with the model simulation if the signal is so strong. 7. L340: Aerosols with radius of 0.385um are easily removed from the convection by

activation.

---

## Referee Comment (RC2) · Anonymous Referee #2 · 14 Oct 2020

Lee et al. studies the convective transport of pollutants from Sichuan basin to Asian Monsoon Anticyclone (AMA) region during one convective event on Aug.7 of 2017. Lee et al. (2020) uses a cloud-chemistry model (Meso-NH) and observational data from the StratoClim, IAGOS and satellites. Lee et al. shows in section 3 that the model reasonably reproduced observed concentrations of some chemical tracers including ozone and CO compared during the Aug.7 convective event. Lee et al. demonstrates using the model that the convection quickly transports CO from boundary layer to 18 km and contributes to 0.5% of CO in the 10-20 km layer for 2 days. Besides, Lee et al. shows that India contributes more than China to the CO in AMA and the Chinese portion is significantly contributed by Sichuan basin. In general, I think the paper reports

an important transport pathway from Sichuan basin to AMA, which is constrained by the StratoClim datasets. However, some concerns are needed to be addressed before publication.

From Figure3, we know that the Aug.7 convective event is reproduced well by the model. In terms of the long-term Chinese/Indian contributions (e.g. 10-day averages in Figure 12), any information to show that the clouds/convections are reasonably simulated during the 10-day period?

For the CNTL and other sensitivity simulations, are the multiple convections similar amount those runs (including the starting time, base height, BT, LWC etc)? I am asking this question because we are talking about 0.5% of anomalies due to Sichuan Aug.7 event. Can we tell the number (i.e. contribution fraction) you derived are statistically significant?

I am concerned by the analysis on aerosol (POA and BC) and Figure 11. What is the parameterization scheme of the convective removal? Does the secondary activation of aerosols (e.g. Grell and Freitas et al., 2014, ACP; Wang et al., 2013, GMD; Yu et al., 2019, GRL) considered in this study? Convection can quickly remove aerosols in-cloud, which results in fast (in log-scale) decay of aerosols From Figure 8, seems modeled POA and BC can be transported from BC to UT without much loss, which seems not right to me. (note, unlike insoluble species CO in your Figure 8, aerosol even BC and POA can be internal mixed and activated).

Minor concerns:

For Meso-NH CNTL run, what are the initial conditions for clouds? Are aerosols activated to CCN in Meso-NH, which can influence the cloud droplet number? Since this study heavily relies on the parameterizations of the convections (which shows pretty nice agreement in Figure 3), more information on the aerosol-cloud interaction schemes are needed in the method section.

Figure 6, the colored circles are extremely difficult to find. Might consider using circles with black boundaries.

Figure 12 caption, AMA region? Altitude info is missing.

---

## Author Response (AR1)

**Convective uplift of pollution from the Sichuan basin into the Asian monsoon anticyclone during the StratoClim aircraft campaign**

**By K. O. Lee et al.**

Reply to the referees' comments

In the following, the comments made by the referees appear in black, while our replies are in red, and the proposed modified text in the typescript is in blue.
* * *
Referee #1 comments
* * *
**Summary general Comments**

This manuscript investigated the convective uplift of pollution from the Sichuan basin into the Asian summer monsoon anticyclone during the StratoClim aircraft campaign in 2017 by simulations with the Meso-NH cloud-chemistry model. After validation with the BT from satellite data and CO and ozone from airborne observations, the simulations are believed to study the impact of Sichuan convection on the AMA composition. Overall, the manuscript shows some interesting results, particularly the role of Sichuan basin as source region of pollution. Some major issues should be addressed before acceptance for publication in ACP.

We appreciate the time and effort you put in this review as well your helpful comments on our paper. We have worked hard to improve the manuscript. Replies to each comment are listed below.

**Major Comments:**

1. How to verify the simulations of aerosol in the tropopause layer? Because no measurements are used to validate the simulation. There are already some published studies, where the observations of aerosol signal are used to validate the simulations.

During the StratoClim aircraft campaign in 2017, UHSAS (ultra high sensitive aerosol spectrometer) was embarked on the Geophysica aircraft to measure the total number concentration of aerosol in size range of 0.65–1000 nm. The measured and simulated particle number concentrations are compared in Figure A. In the figure, the particle number concentrations observed by UHSAS during flights #F6 (a), #F7 (b), and #F8 (c) are displayed by red marks while the simulated domain-averaged number concentration within a box covering each flight track by Meso-NH control run is displayed by black solid line.

[Figure]

**Figure A.** Profiles of particle number concentration measured by UHSAS sensor (red crosses marks) aboard flights (a) #F6, (b) #F7, and (c) #F8 and simulated by the Meso-NH model in a box area covering each flight tracks (black solid line).

Figure A shows that the model captures the general profile shape of particle number concentration in the 0–400 range in the altitudes above 4 km as measured. However the model missed the large number concentration stored in lower troposphere below 4 km. Also, we also have performed the apple to apple comparison (not shown) as we did for CO and $O_3$ (Figures 4, 5, and 6), however model couldn't capture the detailed variation of aerosol along the flight tracks. This discrepancy may be linked to the accuracy of the initial forcing (i.e. MOZART), coarse model grid spacing (i.e. 15 km), normalized aerosol size and number distributions at initial state, and/or etc. This part will be more deeply investigated in future study to produce more realistic aerosol distribution. However the purpose here is to understand the lifecycles of primary pollutants, CO, primary organic aerosol (POA) and black carbon (BC) uplifted in the Sichuan basin by deep convection after thorough validation using airborne in situ measurements.

2. Could you validate the CO simulations with MLS data? The simulations are not so good for F6 and F8 as shown in Fig. 4, so more validations are in need. MLS CO data will surely show the enhanced CO plume after the deep convection if the results given in this paper are correct.
The simulation for F6 (Fig. 4b) couldn't capture the short 2 observed-CO peaks during 32000–36000 s, but rather smooth CO enhancements. However, except for this 4000 s period, the model captures very well most of the variations. For F8 (Fig. 4d), the model underestimates the CO concentration during the first leg (32000–34000 s) and last leg (42000–44300 s), and it misses short CO variations occurring at 35000 s, 35400 s, and 37500 s. Nevertheless, the model reproduces the general CO variations during F8.

Even if the model failed to capture some detailed CO features, it successfully captured the general CO variations during F6 and F8. It is also important to mention that the agreement is even better for F5 and F7. And, concerning F7 the model reproduces CO enhancements clearly identified as Sichuan contributions by the sensitivity tests performed with the model.

Such model-observation agreement is therefore largely enough to validate the ability of the model to document the impact of overshooting convection on the AMA composition. The missed short CO peaks for F6 and are probably linked to the coarse model horizontal resolution not adapted to capture the finest plumes while the overestimated CO concentrations during take-off and landing may be related to the emission inventory The Above discussion has been further strengthened in the manuscript.

♣ Page 8, line 231–237
"[…] As for flight #6 the model is not able to reproduce the short CO peaks but instead produces longer and smoother increases. For flight #8, the model missed the very short CO peaks at ~17 km. This is probably linked to a too coarse model grid spacing not adapted to capture fine CO plumes. For F8, Aat the beginning and end of the flight, the aircraft ascends to 19.4 km and the simulation overestimates the concentrations by up to 30 ppb. About 40 ppb overestimation which is also clear for the tropospheric profiles during take-off and landing. This is probably linked to the emission inventory."

As mentioned by referee, MLS has been widely used for studies in the upper troposphere and lower stratosphere (UTLS). In particular, it has been used to document large scale variations of CO in the AMA and to validate UTLS CO distributions from global scale models (Park et al., 2007; Park et al., 2009; Barret et al., 2008). Nevertheless, MLS has a limited spatio-temporal coverage (3500 profiles a day) and can hardly resolve fine scale (< 100km; < 2–5 days) CO structures such as those reproduced by our simulations. Furthermore, in August and September 2017 (d213–d273) MLS data are not available on the site official MLS site (https://disc.gsfc.nasa.gov/datasets/). Thus authors kindly propose to keep Figure 4 as it is.

**Minor Comments:**

1. L43–45: A recent paper by Bian et al. (2020) gives a comprehensive review of the deep convection on the UTLS composition during the ASM, which is recommended to be cited here. Bian, J. et al. 2020: Transport of Asian surface pollutants to the global stratosphere from the Tibetan Plateau region during the Asian summer monsoon, National Science Review, 7, 516–533, doi:10.1093/nsr/nwaa005.

Thank you for suggesting the recent article. This has been cited in the manuscript.

♣ Page 2, line 44–45

"[…] have a significant chemical and radiative impact (Mason and Anderson, 1963; Dickerson et al., 1987; Randel and Park, 2006; Su et al., 2011; Fadnavis et al., 2013; Gu et al., 2016; Bian et al., 2020). [...]"

♣ Page 17, line 508–510

Bian, J., Li, D., Bai, Z., Li, Q., Lyu, D., and Zhou, X.: Transport of Asian surface pollutants to the global stratosphere from the Tibetan Plateau region during the Asian summer monsoon. National Science Review, 7, 516–533, doi:10.1093/nsr/nwaa005, 2020.

2. L52: Higher tropopause over the ASM region is shown by Bian et al. (2012), which also shows the structure of AMA and therefore is recommended to be cited here. Bian, J., L. L. Pan, L. Paulik, H. Vomel, H. Chen, and D. Lu, 2012: In situ water vapor and ozone measurements in Lhasa and Kunming during the Asian summer monsoon. Geophys. Res. Lett., 39, L19808, doi:10.1029/2012GL052996.

Thank you again. This has been cited in the manuscript.

♣ Page 2, line 52–54

"[…] above the ASM is relatively high (16–17.5 km) and the AMA extends into the lower stratosphere spanning from around 200 hPa to 70 hPa (12–18.5 km above sea level), i.e. approximately the whole UTLS (Highwood and Hoskins, 1998; Randel and Park, 2006; Bian et al., 2012)."

♣ Page 16, line 511–512

Bian, J., Pan, L.L., Paulik, L., Vomel, H., Chen, H., and Lu, D.: In situ water vapor and ozone measurements in Lhasa and Kunming during the Asian summer monsoon. Geophys. Res. Lett., 39, L19808, doi:10.1029/2012GL052996, 2012.

3. L64–68: The different contribution from Indian and China sources to the UTLS is investigated by Yan et al. (2015), which conducts the simulation for one month with WRF-chem model and is recommended to be cited here. Yan, R. and J. Bian, 2015: Tracing the boundary layer sources of carbon monoxide in the Asian summer monsoon anticyclone using WRF-Chem. Adv. Atmos. Sci., 32(7), 943–951, doi:10.1007/s00376-014-4130-3.

This has been cited in the manuscript.

♣ Page 2, line 64–65

"[…] demonstrated that the BL pollution uplifted to the AMA was mostly from Indian or South Asian sources (Park et al., 2009; Yan and Bian, 2015; Barret et al., 2016). [...]"

♣ Page 22, line 685–686

Yan, R. and J. Bian: Tracing the boundary layer sources of carbon monoxide in the Asian summer monsoon anticyclone using WRF-Chem. Adv. Atmos. Sci., 32(7), 943–951, doi:10.1007/s00376-014-4130-3, 2015.

4. L180–190: How are the CCN activation and second activation by entrainment in the convective cloud considered which is critical to the simulation of aerosol profile as shown by Yu et al. (2018)? Yu, P., K.D. Froyd, R.W. Portmann, O.B. Toon, S. R. Freitas, C. G. Bardeen, C. Brock, T. Fan, R. S. Gao, J. M. Katich, A. Kupc, S. Liu, C. Maloney, D. M. Murphy, K. H. Rosenlof, G. Schill, J. P. Schwarz and C. Williamson (2019), Efficient In-cloud Removal of Aerosols by Deep Convection, Geophys. Res. Lett., 45, 1061-1069, https://doi.org/10.1029/2018GL080544

We do not use CCN activation and second activation in this simulation. There is no great physical sense in using aerosol activation parameterization at the horizontal resolution of the model (i.e. 15 km) without being able to compute supersaturation in the air parcel. The model is not in cloud-resolving-model (CRM) configuration. We therefore use a microphysical scheme at a time well adapted to this scale (ICE3, Pinty and Jabouille, 1998). This scheme follows the approach of Lin et al. (1983) in that a three-class ice parameterization is coupled to a Kessler's scheme for warm processes. The convection scheme is Kain-Fritch-Bechtold (Bechtold et al., 2001) also widely used by the international community and well adapted to this resolution. Moreover the employed scheme is able to show the convective uplift of pollutant from boundary layer to the UTLS via deep convective cloud. To the sake of clarity about cloud and aerosol interaction, more information has been included in the manuscript.

♣ Page 6, lines 170–175

"[...] Tulet et al., 2005, 2006, 2010). The impaction scavenging by raindrops depends mainly on Brownian motion, interception, and inertial impaction following a formula originally described by Slinn (1983). Two lognormal modes of particles are considered, mode #1 (i.e. Aitken mode) of smaller particles with initial mean radius of 0.036 μm and standard deviation (σ) of 1.86, and mode #2 (i.e. accumulation mode) of larger particle with initial mean radius of 0.385 μm and σ of 1.29. The coarse mode of the particles is strongly leached by impaction, while the Aitken and nucleation modes are collected by Brownian motion. The gas to particle [...]"

♣ Page 6, lines 187–192

"[...] each condensed water species has a nonzero fall speed. In this study, Meso-NH simulation have a horizontal grid spacing of 15 km with parameterized convection resulting from a trade-off between a high resolution for detailed dynamics of the mesoscale convective systems an efficient run over a large domain covering the entire AMA. There is certainly an effect of not explicitly considering aerosol activation on clouds that is difficult to quantify without performing a higher resolution simulation. However, in deep convection, high vertical velocities create significant supersaturation and tend to activate much of the available aerosol spectrum. The turbulence parameterisation is based on a 1.5-order closure [...]"

5. Fig. 6 and Fig. 7: CO data from MLS is suggested to compare with the model simulation.
Please see our reply to major comment #2.

6. Fig. 10: CALIPSO data for aerosol is suggested to compare with the model simulation if the signal is so strong.
Indeed vertical profiles of backscatter retrieved from the Cloud-Aerosol Lidar with Orthogonal Polarization (CALIOP) on board CALIPSO (Winker et al. 2009) with a wavelength at 532 nm has been widely used to understand the particle contribution. CALIOP data has been thus employed in many studies regarding of not only aerosol (Mielonen et al., 2009; Yorks et al., 2009; Devasthale and Thomas, 2011), but also hydrometeors, i.e. ice crystals in high clouds (Yoshida et al., 2010; Baum et al., 2011; Lee et al., 2019).

StratoClim campaign took place during a break phase of the monsoon with an intense convective activity over south China. Many deep convective clouds transported a large amount of both solid hydrometeor and boundary aerosol to ATAL, thus the ATAL composition during StratoClim campaign is a mixture of both. Actually using CALIOP data, the stratospheric hydration by deep convections during flight #7 had been studied by Lee et al. (2019). They reported that ice content up to 1.9 eq. ppmv distributes in altitudes of 17–18 km in the upper troposphere. Figure A (part of Figure 3 in Lee et al., 2019) demonstrates the V-shaped high backscatter signal region in altitudes of 16–18 km (pointed by a white arrow in Fig. Aa) are successfully reproduced by Meso-NH simulation (Fig. Ab) as observed by CALIOP. Furthermore the V-shaped region is mostly composed with ice contents (Fig. Ac). Their study shows that during active convection phase of summer monsoon, strong backscatter signals in high altitudes detected by CALIOP might not point out only aerosol. Thus we prefer not to include CALIOP data in Figure 10.

[Figure]

**Figure A.** Backscatters at 532 nm (a) measured by CALIOP around 20:00 UTC and (b) retrieved by the Meso-NH simulation, and ice content (eq. ppmv) produced by the Meso-NH simulation along the CALIOP track at 20:00 UTC on 7 August 2017. (Lee et al., 2019)

7. L340: Aerosols with radius of 0.385 μm are easily removed from the convection by activation.
You are right. For the sake of clarity, additional information has been included in the manuscript.

♣ Page 6, lines 170–175
"[…] Tulet et al., 2005, 2006, 2010). The impaction scavenging by raindrops depends mainly on Brownian motion, interception, and inertial impaction following a formula originally described by Slinn (1983). Two lognormal modes of particles are considered, mode #1 (i.e. Aitken mode) of smaller particles with initial mean radius of 0.036 μm and standard deviation (σ) of 1.86, and mode #2 (i.e. accumulation mode) of larger particle with initial mean radius of 0.385 μm and σ of 1.29. The coarse mode of the particles is strongly leached by impaction, while the Aitken and nucleation modes are collected by Brownian motion. The gas to particle [...]"

**General Comments**

Lee et al. studies the convective transport of pollutants from Sichuan basin to Asian Monsoon Anticyclone (AMA) region during one convective event on Aug. 7 of 2017. Lee et al. (2020) uses a cloud-chemistry model (Meso-NH) and observational data from the StratoClim, IAGOS and satellites. Lee et al. shows in section 3 that the model reasonably reproduced observed concentrations of some chemical tracers including ozone and CO compared during the Aug. 7 convective event. Lee et al. demonstrates using the model that the convection quickly transports CO from boundary layer to 18 km and contributes to 0.5% of CO in the 10-20 km layer for 2 days. Besides, Lee et al. shows that India contributes more than China to the CO in AMA and the Chinese portion is significantly contributed by Sichuan basin. In general, I think the paper reports an important transport pathway from Sichuan basin to AMA, which is constrained by the StratoClim datasets. However, some concerns are needed to be addressed before publication.

We appreciate the time and effort you put in this review as well your helpful comments on our paper. We have worked hard to improve the manuscript. Replies to each comment are listed below.

**Major Comments:**

1. From Figure 3, we know that the Aug. 7 convective event it reproduced well by the model. In terms of the long-term Chinese/Indian contributions (e.g. 10-days averages in Figure 12), is there any information to show that the clouds/convections are reasonably simulated during the 10-day period?

For the sake of clarity on the long-term model ability to reproduce convective clouds, we have joined here the 12-hourly observed and simulated images of brightness temperature (BT, unit in K) from 1st to 10th August. Figure A shows the composite BT images using SEVIRI/MSG and Himawari, and Figure B shows the simulated BT images using CNTL run.

   The figures demonstrate that the spatial coincidence of clouds and deep convection (BT ≤ 210 K) is globally good during the 10-days period. Also it shows that the lifecycle of convective clouds within the ASM (Asian Summer Monsoon, south and East Asia from the tropics to the subtropics) circulation is reasonably reproduced by the model. Compared to observed images, Meso-NH tends to slightly underestimate the horizontal extension and the intensity of convective clouds. This piece of information has been included in the manuscript.

♣ Page 13, lines 415–416

"source regions. Observed and simulated clouds are globally coincident during the 1–10 August period. The model slightly underestimates their extension and intensity. CO distribution [...]"

[Figure]

**Figure A.** BT composite images using SEVIRI/MSG and Himawari at (a) 06:00 UTC, (b) 18:00 UTC on 1 August, (c) 06:00 UTC, (d) 18:00 UTC on 2 August, (e) 06:00 UTC, (f) 18:00 UTC on 3 August, (g) 06:00 UTC, (h) 18:00 UTC on 4 August, (i) 06:00 UTC, (j) 18:00 UTC on 5 August, (k) 06:00 UTC, (l) 18:00 UTC on 6 August, (m) 06:00 UTC, (n) 18:00 UTC on 7 August, (o) 06:00 UTC, (p) 18:00 UTC on 8 August, (q) 06:00 UTC, (r) 18:00 UTC on 9 August, (s) 06:00 UTC and (t) 18:00 UTC on 10 August 2017. The coastlines are marked by black solid lines.

[Figure]

**Figure B.** Same as Figure A but from the Meso-NH simulation.

2. For the CNTL and other sensitivity simulations, is the multiple convections similar amount those runs (including the starting time, base height, BT, LWC etc)? I am asking this question because we are talking about 0.5% of anomalies due to Sichuan Aug. 7 event. Can we tell the number (i.e. contribution fraction) you derived are statistically significant?

Convection in the CNTL and 4 sensitivity simulations have been identically initialized with ECMWF analyses and identically parameterized with the Kain-Fritsch-Bechtold scheme (Bechtold et al., 2001). The simulated convective systems are thus exactly identical, in terms of lifetime, depth, LWC, etc. For instance, Figure C shows the BT distributions reproduced by the CNTL and the four sensitivity simulations at 18:00 UTC on 7 August, time of matured deep convection over the Sichuan Basin. The figure demonstrates that the horizontal extent and depth of deep and shallow clouds reproduced by CNTL and sensitivity simulations are exactly the same. This is true for the other meteorological variables such as humidity or temperature.

The only change among the five simulations concerns the surface emissions. From the SIC06 simulation we determined that Sichuan pollution convectively uplifted by the 7 August event was responsible of large and significant CO enhancements (6 to 12 %) over a 1000 km broad region (Figure 7i). The 0.5 % contribution is given to provide an idea about the impact of pollution uplifted by the 7 August convective event over the whole AMA region. It may seem small because of the dilution effect but it is as significant as the enhancement over the Sichuan region. Moreover, this Sichuan CO contribution is still detectable over Nepal as confirmed by the StratoClim observation during F7 (Figure 4c). Figure 9 also shows that this 0.5% contribution remains steadily until 9 August 12:00 UTC in the two uppermost layers (10–20 km).

[Figure]

**Figure C.** Identical BT (K) distributions at 18:00 UTC on 7 August from (a) CNTL, (b) SIC06, (c) SIC01, (d) CHN01, and (e) IND01 simulations.

♣ Page 9, lines 290–291

"[…] All the other environmental conditions are identical to the CNTL run, and the convection activities (i.e. lifetime, intensity) between simulations are as well identical."

♣ Page 12, lines 369–370

"[…] until 00:00 UTC on 8 August. Note the convective activity is identical in all experiments. […]"

3. I am concerned by the analysis on aerosol (POA and BC) and Figure 11. What is the parameterization scheme of the convective removal? Does the secondary activation of aerosols (e.g. Grell and Freitas et al., 2014, ACP; Wang et al., 2013, GMD; Yu et al., 2019, GRL) consider in this study? Convection can quickly remove aerosols in-cloud, which results in fast (in log-scale) decay of aerosols from Figure 8, seems modeled POA and BC can be transported from BC to UT without much loss, which seems not right to me. (note, unlike insoluble species CO in your Figure 8, aerosol even BC and POA can be internal mixed and activated).

The wet deposition scheme is based on Tulet et al. (2010). The kinetic mass transfer between aerosols and cloud or rain drops is considered. The impaction scavenging by raindrops depends mainly on Brownian motion, interception, and inertial impaction following a formula originally described by Slinn (1983). See also Seinfeld and Pandis (1997), Pruppacher and Klett (2000), Tost et al. (2006) for classical parameterization in mesoscale models. Thus the collection efficiency depends on the size of the aerosols (and secondarily on the size of the raindrops). The coarse mode of the particles is strongly leached by impaction. The Aitkin and nucleation modes are collected by Brownian motion.

On the other hand, the accumulation mode is globally little impacted by these two processes and remains preserved in the cloud. This is physically true for insoluble aerosols such as BC, which are not CCN. So the reviewer is right to say that if BC becomes hygroscopic by mixing with soluble secondary compounds (organic for example), it becomes potentially CCN and should be activated into clouds droplets. This process is not taken into account in the simulation and is a source of error. This limitation has been mentioned in the text.

♣ Page 6, lines 170–175

"[…] Tulet et al., 2005, 2006, 2010). The impaction scavenging by raindrops depends mainly on Brownian motion, interception, and inertial impaction following a formula originally described by Slinn (1983). Two lognormal modes of particles are considered, mode #1 (i.e. Aitken mode) of smaller particles with initial mean radius of 0.036 µm and standard deviation (σ) of 1.86, and mode #2 (i.e. accumulation mode) of larger particle with initial mean radius of 0.385 µm and σ of 1.29. The coarse mode of the particles is strongly leached by impaction, while the Aitken and nucleation modes are collected by Brownian motion. The gas to particle [...]"

♣ Page 11, lines 355–359

"[…] This result thus implies that aerosol sizes in both modes within the polluted plume are increased during the uplifting within the cloud by gas-particles conversion, condensation of water in the aerosol and coagulation (Andronache, 2003; Tost et al., 2007; Berthet et al., 2010; Tulet et al., 2010). Note that mixing of insoluble aerosols such as BC with soluble secondary compounds to become hygroscopicand potentially CCN (cloud condensate nuclei) that could be activated into cloud droplets is not taken into account in the simulation. […]"

**Minor Comments:**

1. For Meso-NH CNTL run, what are the initial conditions for clouds?

Clouds are formed after a saturation adjustment. The model is initialized by ECMWF analyses and the cloud formation will therefore take place in the first time steps of the model (spin-up). Generally, the model is well balanced after 2 hours of simulation. This piece of information has been included in the manuscript.

♣ Page 6, lines 182–183

"The meteorological conditions are initialized by the ECMWF analyses and clouds are formed in the first time steps of the model (spin-up) after a saturation adjustment. Deep convection is parameterised […]"

2. Are aerosols activated to CCN in Meso-NH, which can influence the cloud droplet number? Since this study heavily relies on the parameterizations of the convections (which shows pretty nice agreement in Figure 3), more information on the aerosol-cloud interaction schemes are needed in the method section.

We do not use aerosol activation in this simulation. There is no great physical sense in using aerosol activation parameterization at the horizontal resolution of the model (i.e. 15 km) without being able to compute supersaturation in the air parcel. At this resolution, convection, a part of the clouds and precipitation are not explicitly resolved. The model is not in cloud-resolving-model (CRM) configuration. We therefore use a microphysical scheme at a time well adapted to this scale (ICE3, Pinty and Jabouille, 1998). This scheme follows the approach of Lin et al. (1983) in that a three-class ice parameterization is coupled to a Kessler's scheme for warm processes. The convection scheme is Kain-Fritch-Bechtold (Bechtold et al., 2001) also widely used by the international community and well adapted to this resolution.

In order to study the effect of aerosol activation on clouds, a grid-nesting simulation should be carried out to reach the resolved cloud scale (< 3 km horizontal resolution). There is certainly an effect of not explicitly considering aerosol activation on clouds that is difficult to quantify without performing a higher resolution simulation. However, in deep convection, high vertical velocities create significant supersaturation and tend to activate much of the available aerosol spectrum (CCN). Thus, and particularly in polluted environments, we can reasonably assume that CCN are not a limiting factor in cloud formation. This has been mentioned in the manuscript.

♣ Page 6, lines 187–192

"[...] each condensed water species has a nonzero fall speed. In this study, Meso-NH simulation have a horizontal grid spacing of 15 km with parameterized convection resulting from a trade-off between a high resolution for detailed dynamics of the mesoscale convective systems an efficient run over a large domain covering the entire AMA. There is certainly an effect of not explicitly considering aerosol activation on clouds that is difficult to quantify without performing a higher resolution simulation. However, in deep convection, high vertical velocities create significant supersaturation and tend to activate much of the available aerosol spectrum. The turbulence parameterisation is based on a 1.5-order closure [...]"

3. Figure 6, the colored circles are extremely difficult to find. Might consider using circles with black boundaries.

Thanks for this suggestion. For the sake of visibility, Figure 6 (a and c) has been improved by closing the circles with black boundaries and reducing the data interval to 5 min from 4 s.

[Figure]

**Figure 6.** IAGOS-measured (dashed lines) and Meso-NH-derived (solid lines) carbon monoxide (black lines) and $O_3$ (blue lines) along IAGOS flight tracks on 5 August 2017. In (a) and (c), Meso-NH-derived CO at the altitude of 11.1 km are displayed by shaded areas, while the IAGOS-measured CO every 5 min are displayed by coloured circles along the track (red lines). In (b) and (d), IAGOS-measured CO and $O_3$ every 4 s are displayed. In (a)–(d), the starting (ending) point of each flight within the domain is marked by open (closed) red circle, while the location of the steep (gradual) change of carbon monoxide is marked by red (black) arrows.

4. Figure 12 caption, AMA region? Altitude info is missing.
Indeed. This piece of information has been included.

[Figure]

**Figure 12.** 10-days averaged chemical components of CO (ppbv) at the altitude of 14.8 km from 1 to 10 August 2017 produced by (a) CNTL, (b) CNTL minus CHN01, (c) CNTL minus SIC01, and (d) CNTL minus IND01.

---

## Referee Report (RR1)

This paper discussed the chemical properties of the South Asian Upper Troposphere Lower Stratosphere during the Asian Summer Monsoon. They used Meso-NH cloud-chemistry model to simulate the monsoon deep convection on the composition of Asian Monsoon Anticyclone at the 15 km spatial resolution during the StratoClim campaign in 2017. The simulated CO, O3, POA, and BC were compared with StratoClim and IAGOS observations. Overall, this paper is good, except for some defects in the analysis and model description.

General comments:

1. The simulation was conducted at 15 km resolution. The Kain-Fritsch-Bechtold scheme was used to simulated the parameterized (subgrid) convection. Do you use any scheme to simulate the subgrid convective transport and wet scavenging of the chemical properties? Or maybe the Kain-Fritsch-Bechtold scheme included the subgrid convective transport and wet scavenging of the chemical properties? Could you show more details about the subgrid convective transport and wet scavenging of the chemical properties in the model description part?

   The subgrid convective transport and wet scavenging of the chemical properties are very important at 15 km spatial resolution especially in the developing stage of the convection. You can refer to the following two papers for more details:

   Grell, G. A., & Freitas, S. R. (2014). A scale and aerosol aware stochastic convective parameterization for weather and air quality modeling. *Atmospheric Chemistry and Physics*, 14(10), 5233–5250. https://doi.org/10.5194/acp-14-5233-2014

   Li, Y., Pickering, K. E., Barth, M. C., Bela, M. M., Cummings, K. A., & Allen, D. J. (2018). Evaluation of parameterized convective transport of trace gases in simulation of storms observed during the DC3 field campaign. *Journal of Geophysical Research: Atmospheres*, 123, 11,238–11,261. https://doi.org/10.1029/2018JD028779

2. The concentrations of the chemical properties in the upper troposphere are very sensitive to the relative location of the storm. Could you plot the flight tracks over the BT plots (or over radar reflectivity, or anything that can show the storm location) for each flight?

Specific comments:
   1. Line 25: What's IAGOS? Could you show the full name of IAGOS?

   2. Line 60: If you decided to use CO to represent carbon monoxide, please keep consistent throughout the paper (e.g in line 102, and in the captions of Figures 2, 4, 6, 7, 9).

   3. Line 62: Please show the full name of CALIPSO.

   4. Line 83: Please show the full name of MACCity.

   5. Line 100: Could you show the aircraft tracks over radar reflectivity or BT to show their relative location to the convections?

6. Line 105: Please show the full name of HITRAN.

7. Line 124: Please show the full name of CTL.

8. Section 2.3: What's the time resolution of the simulation? It would be better if you can use a table to show the model setups.

9. Line 181: Are these meteorology schemes are the best combination? Did you try other schemes?

10. Line 181: How did the model transport the chemical properties in the subgrid scale? See general comments 1 for more details.

11. Section 3.2: Could you compare the simulation with the observation before convection started to prove the accuracy of the initial chemistry condition?

12. Section 3.2: When you compared the simulated and observed chemical properties, did you use a criterion to separate the inside-cloud and outside-cloud region? The CO concentration might be very different inside and outside the cloud.

13. Line 265: The height of the tropopause layer might increase in the area of deep convections. Therefore, it is not good to the climatological TTL height to determine whether the pollution affected the lower stratosphere or not. It's better to use the temperature gradient (like the WMO tropopause definition) to determine the height of the TTL.

14. Line 290: How do you define the cloud boundary? I see the definition in the caption of Figure 8, could you also mention it in the main content of the paper?

15. Line 321: Could you use an equation to describe the contribution of Sichuan?

16. Line 324: Change "The evolution of the difference …" to "The evolution of the contribution of Sichuan …"

17. Figure 1. What's STCLM?

18. Figure 2: This is the first time "CNTL", "SIC06", "SIC01", "CHN01", and "IND01" were mentioned in the paper. You may need to explain the meaning of these abbreviations.

19. Figure 7: Change "Horizontal map of BT …" to "Horizontal map of simulated BT …"

20. Figure 9: Please add the label of the x-axis (i.e. Date).

21. Figure 9: Please show the description of the blue box in the legend and caption.

---

## Author Response (AR2)

**Convective uplift of pollution from the Sichuan basin into the Asian monsoon anticyclone during the StratoClim aircraft campaign**

**By K. O. Lee et al.**

Reply to the referees' comments

In the following, the comments made by the referees appear in black, while our replies are in red, and the proposed modified text in the typescript is in blue.
* * *
Referee #1 comments
* * *
**Comments**

The revised version has answered most of my concerns. My only concern is the simulation of the aerosol in the upper troposphere and lower stratosphere, which has no support from measurement and is not the critical concern of this manuscript. My suggestion is to remove this part.

We appreciate the time and effort you put in this review as well your helpful comments on our paper. In this study, we only discuss and display primary particles POA and BC in order to compare their distributions with a primary gas, CO. This allows to show that these particles have a different fate and different distributions due to their interactions with clouds which is not the case of CO. It is true that the part regarding the simulation of aerosol, i.e. primary particles POA and BC, in the upper troposphere and lower stratosphere (UTLS) has not been supported by measurement. However authors believe that the model results shows interesting characteristics of primary particles with respect to the development of deep convective system, and the need of future field campaigns with innovative instrument which can detect various aerosol particles. With these reason, we kindly propose to keep the part about the simulation of aerosol as it is, while further discussion has been included as below.

♣ Page 12, lines 378–381

"[…] stored at the mountain foothills into the AMA. The simulation results shows us interesting aspects of primary particles of POA and BC with respect to the development of deep convective clouds however it is not supported by measurement and show the need of future field campaigns deploying instrument which can detect various aerosol particles."
* * *
Referee #2 comments
* * *
**Summary general Comments**

This paper discussed the chemical properties of the South Asian Upper Troposphere Lower Stratosphere during the Asian Summer Monsoon. They used Meso-NH cloud-chemistry model to simulate the monsoon deep convection on the composition of Asian Monsoon Anticyclone at the 15 km spatial resolution during the StratoClim Campaign in 2017. The simulated CO, O3, POA, and BC were compared with StratoClim and IAGOS observations. Overall, this paper is good, except for some defects in the analysis and model description.

We appreciate the time and effort you put in this review as well your helpful comments on our paper. We have worked hard to improve the manuscript. Replies to each comment are listed below.

**General Comments:**

1. The simulation was conducted at 15 km resolution. The Kain-Fritsch-Bechtold scheme was used to simulate the parameterized (subgrid) convection. Do you use any scheme to simulate the subgrid convective transport and wet scavenging of the chemical properties? Or maybe the Kain-Fritsch-Bechtold scheme included the subgrid convective transport and wet scavenging of the chemical properties? Could you show more details about the subgrid convective transport and wet scavenging of the chemical properties in the model description part?

The subgrid convective transport and wet scavenging of the chemical properties are very important at 15 km spatial resolution especially in the developing stage of the convection. You can refer to the following two papers for more details:

Grell, G.A., and Freitas, S.R. (2014). A scale and aerosol aware stochastic convective parameterization for weather and air quality modeling. Atmospheric Chemistry and Physics, 14(10), 5233–5250. https://doi.org/10.5194/acp-14-5233-2014.

Li, Y., Pickering, K.E., Barth, M.C., Bela, M.M., Cummings, K.A., and Allen, D.J. (2018). Evaluation of parameterized convective transport of trace gases in simulation of storms observed during the DC3 field campaign. Journal of Geophysical Research: Atmospheres, 123, 11238–11261. https://doi.org/10.1029/2018JD028779.

As mentioned in previous studies, e.g. Grell and Freitas (2014) and Li et al. (2018), to obtain reasonable simulations of the impact of convection on tropospheric composition, subgrid convective transport needs to be computed in a manner consistent with the subgrid convection in the driving meteorological model. In this study, the Kain-Fritsch-Bechtold scheme (Bechtold et al., 2001) was used with a horizontal grid spacing of 15 km. Scavenging by subgrid wet convective updrafts is applied within the convective mass transport algorithm in order to prevent soluble tracers from being transported to the top of the convective updraft and then dispersed on the grid scale. The transport model provides wet convective air mass fluxes through each grid level in the updraft. This piece of information has been included in the model description part (section 2.3).

♣ Page 6, lines 194–198

"[…] et al., 2001). In order to obtain reasonable simulations of the impact of convection on tropospheric composition, subgrid convective transport needs to be computed (Grell and Freitas, 2014; Li et al., 2018). Scavenging by subgrid wet convective updrafts is applied within the convective mass transport algorithm in order to prevent soluble tracers from being transported to the top of the convective updraft and then dispersed on the grid scale. The transport model provides wet convective air mass fluxes through each grid level in the updraft. […]"

♣ References

Grell, G.A. and Freitas, S.R.: A scale and aerosol aware stochastic convective parameterization for weather and air quality modelling. Atmospheric Chemistry and Physics, 14(10), 5233–5250. doi:105194/acp-14-5233-2014, 2014.

Li, Y., Pickering, K.E., Barth, M.C., Bela, M.M., Cummings, K.A., and Allen, D.J.: Evaluation of parameterized convective transport of trace gases in simulation of storms observed during the DC3 field campaign. Journal of Geophysical Research: Atmosphere, 123, 11238–11261. doi:10.1029.2018JD028779, 2018.

2. The concentrations of the chemical properties in the upper troposphere are very sensitive to the relative location of the storm. Could you plot the flight the flight tracks over the BT plots (or over radar reflectivity, or anything that can show the storm location) for each flight?

Agreed. Figure 1 has been extended by including four sub-figures. The StratoClim flight track of #5, #6, #7, and #8 are displayed in (b), (c), (d), and (e), respectively, on the BT distribution at the time around flight departure. Manuscript and caption of Figure 1 have been accordingly revised as below:

♣ Page 4, lines 102–108

"We will use data from M55-Geophysica flights #5, #6, #7, and #8, which took place from Kathmandu in Nepal (Table 1; for the tracks see red bluish lines marked by 'STCLM' in Fig. 1a and green lines in Fig. 1b–1e). During those flights, the AMICA (Airborne Mid Infrared Cavity enhanced Absorption spectrometer) and FOZAN-II (Fast OZone ANalyzer) instruments measured the CO and $O_3$ concentrations respectively. During 03:00–07:25 UTC on 4 August for flight #5 (Fig. 1b), during 07:30–11:30 UTC on 6 August for flight #6 (Fig. 1c), during 04:30–06:50 UTC on 8 August for flight #7 (Fig. 1d), and during 08:40–12:30 UTC on 10 August for flight #8 (Fig. 1e). Flights #5 and #7 flied in the region almost without tall clouds while flights #6 and #8 flied in the cloudy neighbours."

♣ Page 8, lines 247–252

"[…] for flight #6 it appears that the model is not able to reproduce the short CO peaks but instead produces longer and smoother increases. For flight #8, the model missed the very short CO peaks at ~17 km. This is probably linked to a too coarse model grid spacing not adapted to capture fine plumes. Also note that flights #6 and #8 flied into populated regions of convective clouds (BT ≤ 210 K; see Figs. 1c, 1e). The location of convective clouds strongly affects the concentrations of chemical properties. […]"

[Figure]

Figure 1. (a) Topography and domain considered in the Meso-NH numerical simulation with a resolution of 15 km. The trajectory of the Geophysica flights #5, #6, #7, and #8 during the StratoClim campaigns (marked with 'STCLM' which is a shortened name of StratoClim) around and south of Kathmandu are shown by the bluish solid lines in (a), while the trajectory is displayed by green solid lines and overlapped on BT distribution at the time around each flight's departure in (b), (c), (d), and (e), respectively. In (a),  two IAGOS flight tracks (to/from Madras in India) are indicted by the red solid and dashed line.

**Specific Comments:**

1. Line 25: What's IAGOS? Could you show the full name of IAGOS?
Done.

♣ Page 1, line 25–26
"[…] and airborne observations (StratoClim and IAGOS (In-service Aircraft for a Global Observing System) […]"

2. Line 60: If you decided to use CO to represent carbon monoxide, please keep consistent throughout the paper (e.g in line 102, and in the captions of Figures 2, 4, 6, 7, 9).
Done.

3. Line 62: Please show the full name of CALIPSO.
Done.

♣ Page 2, line 66–67

"[…] observations from the CALIPSO (Cloud-Aerosol Lidar and Infrared Pathfinder Satellite Observation) spaceborne lidar evidenced [...]"

4. Line 83: Please show the full name of MACCity.
Done.

♣ Page 3, line 88

"emissions from the MACCity (MACC/CityZEN EU project) inventory (https://eccad3.sedoo.fr/) [...]"

5. Line 100: Could you show the aircraft tracks over radar reflectivity or BT to show their relative location to the convections?
Done. See our answer to major comment #2.

6. Line 105: Please show the full name of HITRAN.
Done.

♣ Page 4, line 112

"[…] parameters taken from the HITRAN (high-resolution transmission molecular absorption) 2012 database […]"

7. Line 124: Please show the full name of CTL.
The *48CTL* is the name of instrument and is not an acronym.

8. Section 2.3: What's the time resolution of the simulation? It would be better if you can use a table to show the model setups.
The time resolution of the simulation had been set to 40 s which is sufficient to simulate the target convective uplift of pollution and to compare it with fine-scale airborne measurement. This piece of information has been included. Also, as suggested, specification of model setup has been summarized in Table 2 as below.

♣ Page 5, line 155

"[…] 7 million grid points; see Table 2 for a detailed description of the model)."

♣ Page 23

**Table 2.** Specification of model setup.

| | |
|---|---|
| Horizontal resolution | 15 km |
| Vertical resolution | 100 m to 450 m |
| Temporal resolution | 40 s |
| Emission | MACCity, MEGAN, FGEDv3 |
| Meteorology boundary | ECMWF analyses |
| Chemical boundary | MOZART-4 |
| Chemical scheme | ReLACS 2 |
| Aerosol modules | ORILAM, ORILAM-SOA |

9. Line 181: Are these meteorology schemes are the best combination? Did you try other schemes?

The combination of the selected meteorological schemes (e.g. microphysical scheme, turbulence parameterization, transport scheme) has already shown its capability to simulate heavy precipitation events in both real and idealized framework (Ducrocq et al. 2008; Bresson et al. 2012; Lee et al. 2018, 2019 among others). Authors are confident with the selected meteorological schemes and it has been mentioned in manuscript.

♣ Page 7, lines 210–213

"[…] (Colella and Woodward, 1984). This combination of meteorological schemes has already shown its capability to simulate heavy precipitation events in both real and idealized frameworks (Ducrocq et al. 2008; Bresson et al. 2012; Lee et al. 2018, 2019 among others). […]"

♣ References

Ducrocq, V., Nuissier, O., Ricard, D., Lebeaupin, C., and Thouvenin, R.: A numerical study of three catastrophic precipitating events over southern France, Mesoscale triggering and stationary factors, Q. J. Roy. Meteor. Soc., 134, 131–145, 2008.

Bresson, E., Ducrocq, V., Nuissier, O., Ricard, D., and De Saint-Aubin C.: Idealized numerical simulations of quasi stationary convective systems over the Northwestern Mediterranean comple terrain, Q. J. Roy. Meteor. Soc., 138, 1751–1763, doi:10.1002/qj.1911, 2012.

Lee, K.-O., Flamant, C., Duffourg, F., Ducrocq, V., and Chaboureau, J.-P.: Impact of upstream moisture structure on a back-building convective precipitation system in south-eastern France during HyMeX IOP 13. Atmos. Chem. Phys., 18, 16845–16862, doi:10.5194/acp-18-16845-2018, 2018.

Lee, K.-O., Dauhut, T., chaboureau, J.-P., Khaykin, S., Krämer, M., and Rolf, C.: Convective hydration in the tropical tropopause layerduring the StratoClim aircraft campaign; pathway of an observed hydration patch. Atmos. Chem. Phys., 19, 11803–11820, doi:10.5194/acp-19-11803-2019, 2019.

10. Line 181: How did the model transport the chemical properties in the subgrid scale? See general comments 1 for more details.

See our answer to major comment #1.

11. Section 3.2: Could you compare the simulation with the observation before convection started to prove the accuracy of the initial chemistry condition?

For the initial aerosol species are taken from MOZART-4 (Model for Ozone and Related chemical Tracers, version 4) had been used. Background chemistry condition had been compared using IAGOS which measured the CO and $O_3$ concentrations and the comparison results (see section 3.2) shows that the model captures most of the general background variabilities within the AMA (Asian Monsoon Anticyclone).

12. Section 3.2: When you compared the simulated and observed chemical properties, did you use a criterion to separate the inside-cloud and outside-cloud region? The CO concentration might be very different inside and outside the cloud.

Agreed. The CO concentration is sensitive to cloud appearance. Comparison between BT at 10.8 microns observed by satellite sensors and simulated by Meso-NH highlights the ability of the model to correctly reproduce the life cycle of convective clouds; this allows us to continue to validate the CO concentration in the UTLS. Flights #6 and #8 indeed flied into populated regions of convective clouds (BT ≤ 210 K; see Figs. 1c, 1e), and the model missed the peak of CO concentrations whereas model performance was relatively better for flights #5 and #7 which flied into rather clear sky. Above discussion has been included in manuscript (also see our answer to major comment #2).

♣ Page 8, lines 247–251

"[…] flight #6 it appears that the model is not able to reproduce the short CO peaks but instead produces longer and smoother increases. For flight #8, the model missed the very short CO peaks at ~17 km. This is probably linked to a too coarse model grid spacing not adapted to capture fine plumes. Also note that flights #6 and #8 flied into populated regions of convective clouds (BT ≤ 210 K; see Figs. 1c, 1e). The location of convective clouds strongly affects the concentrations of chemical properties. […]"

13. Line 265: The height of the tropopause layer might increase in the area of deep convections. Therefore, it is not good to the climatological TTL height to determine whether the pollution affected the lower stratosphere or not. It's better to use the temperature gradient (like the WMO tropopause definition) to determine the height of the TTL.

Agreed. There exist several tropopause definitions in various literature (e.g. WMO, 1957; Maddox and Mullendore, 2018), considering temperature lapse rate, potential vorticity, static stability, and tracer chemicals. As you mentioned, it is true that there is a difficulty of defining a tropopause in convective environment. The term of TTL has been replaced by UTLS.

♣ Page 9, line 285
"[…] tend to overestimate CO by up to 20 ppbv in the UTLS  especially […].

Maddox, E.M. and G.L. Mullendor: Determination of Best Tropopause Definition for Convective Transport Studies. J. Atmos. Sci., 75, 3433–3446, https://doi.org/10.1175/JAS-D-18-0032.1, 2018.
WMO: Definition of the tropopause, WMO Bull., 6, 136, 1957.

14. Line 290: How do you define the cloud boundary? I see the definition in the caption of Figure 8, could you mention it in the main content of the paper?
Done.

♣ Page 10, lines 309–310
"[…] together with cloud contour (mixing ratio of ice content 10 mg kg$^{-1}$) corresponding to the locations […]"

15. Line 321: Could you use an equation to describe the contribution of Sichuan?
Done.

♣ Page 11, line 345–346
"relative difference between CNTL and SIC06 simulations (= [CNTL minus SIC06] over [CNTL]) averaged over the AMA domain (20–35°N, 60–120°E). […]"

16. Line 324: Change "The evolution of the difference…" to "The evolution of the contribution of Sichuan…"
Done.

♣ Page 11, line 346
"[…] (20–35°N, 60–120°E). The evolution of the contribution of Sichuan  is displayed in Fig. 9 with […]"

17. Figure 1. What's STCLM?
Done.

♣ Page 24
"**Figure 1.** […] the StratoClim campaigns (marked with 'STCLM' which is a shortened name of StratoClim) around […]"

18. Figure 2: This is the first time "CNTL", "SIC06", "SIC01", "CHN01", and "IND01" were mentioned in the paper. You may need to explain the meaning of these abbreviations.
This piece of information has been included in caption of Figure 2.

♣ Page 25
"**Figure 2.** The emission map of CO  used for CNTL (control run). Inner boxes indicate the domain of no emission for sensitivity experiments (see also Table 3) of SIC06 and SIC01 (101–109°E, 26–33°N, red line), CHN01 (100–122°E, 20–40°N, black line), and IND01 (70–95°E, 10–35°N, green line) simulations. For the name of sensitivity experiments, first three alphabets stand for the area of interest, e.g. SIC for Sichuan, CHN for China, and IND for India while the last two numbers stands for the first day of emission modification. For more details, see Table 3."

19. Figure 7: Change "Horizontal map of BT…" to "Horizontal map of simulated BT…"
Done.

"**Figure 7.** Horizontal map of simulated BT […]"

20. Figure 9: Please add the label of the x-axis (i.e. Date).
Done.

[Figure]

**Figure 9.** Temporal evolution of contribution of Sichuan emission to CO  concentration of entire AMA region (60–120°E, 20–35°N) every 5 km from 0.1 km to 20 km altitude from 6 to 9 August 2017. The area of low brightness temperature (210 K) in the Sichuan Basin (101–109°E, 26–33°N) are displayed by blue bar.

21. Figure 9: Please show the description of the blue box in the legend and caption.
Done. See our reply to the comment #20.